# Transcriptomic profiling in canines and humans reveals cancer specific gene modules and biological mechanisms common to both species

**Gregory J. Tawa**[1]*, **John Braisted**[1], **David Gerhold**[1], **Gurmit Grewal**[1], **Christina Mazcko**[2], **Matthew Breen**[3], **Gurusingham Sittampalam**[1], **Amy K. LeBlanc**[2]

**1** National Institutes of Health, National Center for Advancing Translational Sciences, Division of Preclinical Innovation, Therapeutic Development Branch, Rockville, Maryland, United States of America, **2** National Institutes of Health, National Cancer Institute, Center for Cancer Research, Comparative Oncology Program, Bethesda, Maryland, United States of America, **3** Department of Molecular Biomedical Sciences, North Carolina State University, College of Veterinary Medicine, Raleigh, North Carolina, United States of America

\* gregory.tawa@nih.gov

**Data Availability Statement:** All RNA-SEQ binary alignment and index files are available via the NCI's Integrated Canine Data Commons (ICDC) at https://

## Abstract

Understanding relationships between spontaneous cancer in companion (pet) canines and humans can facilitate biomarker and drug development in both species. Towards this end we developed an experimental-bioinformatic protocol that analyzes canine transcriptomics data in the context of existing human data to evaluate comparative relevance of canine to human cancer. We used this protocol to characterize five canine cancers: melanoma, osteosarcoma, pulmonary carcinoma, B- and T-cell lymphoma, in 60 dogs. We applied an unsupervised, iterative clustering method that yielded five co-expression modules and found that each cancer exhibited a unique module expression profile. We constructed cancer models based on the co-expression modules and used the models to successfully classify the canine data. These canine-derived models also successfully classified human tumors representing the same cancers, indicating shared cancer biology between canines and humans. Annotation of the module genes identified cancer specific pathways relevant to cells-of-origin and tumor biology. For example, annotations associated with melanin production (*PMEL*, *GPNMB*, and *BACE2*), synthesis of bone material (*COL5A2*, *COL6A3*, and *COL12A1*), synthesis of pulmonary surfactant (*CTSH*, *LPCAT1*, and *NAPSA*), ribosomal proteins (*RPL8*, *RPS7*, and *RPLP0*), and epigenetic regulation (*EDEM1*, *PTK2B*, and *JAK1*) were unique to melanoma, osteosarcoma, pulmonary carcinoma, B- and T-cell lymphoma, respectively. In total, 152 biomarker candidates were selected from highly expressing modules for each cancer type. Many of these biomarker candidates are under-explored as drug discovery targets and warrant further study. The demonstrated transferability of classification models from canines to humans enforces the idea that tumor biology, biomarker targets, and associated therapeutics, discovered in canines, may translate to human medicine.

caninecommons.cancer.gov, accession number = ICDC000002.

**Funding:** This work was supported (GJT, JB, DG, GG, GS) by the Intramural Research Programs of the National Center for Advancing Translational Sciences, NIH (Z01-TR000249, ncats.nih.gov) and in the case of AKL and CM by the Intramural Program of the National Cancer Institute, NIH (Z01-BC006161, cancer.gov). The funders had no role in study design, data collection and analysis, decision to publish, or preparation of the manuscript.

**Competing interests:** The authors have declared that no competing interests exist.

## Author summary

Understanding relationships between spontaneous cancer in companion (pet) canines and humans can facilitate therapeutic development in both species. Towards this end we developed a protocol that analyzes canine transcriptomics data in the context of existing human data to evaluate comparative relevance of canine to human cancer. We used this protocol to characterize five canine cancers: melanoma, osteosarcoma, pulmonary carcinoma, B- and T-cell lymphoma, in 60 dogs. We identified five gene co-expression modules and found that their pattern of activation was distinct for each cancer. We constructed a classification model based on these modules and used it to successfully classify canine and human transcriptomic data associated with the 5 cancers studied. Annotation of the module genes identified cancer specific pathways relevant to cells-of-origin and tumor biology. The analysis interprets cancer as a species shared, activation pattern, of gene modules each representing different units of biology that come into play for each cancer. 152 biomarker candidates, many of which are understudied as drug targets, were identified for further study. The demonstrated transferability of classification models from canines to humans enforces the idea that tumor biology, biomarker genes and therapeutics derived therefrom, in canines may translate to humans.

## Introduction

The recognition of spontaneous cancer development in companion animals and the potential for inclusion of such animals in biomarker and drug development studies, stems from decades of scientific observation that pet dogs spontaneously develop malignancies that share morphologic, histologic, and biologic characteristics with human cancers. [1,2] Human-canine similarities include shared environment, intact host immunity, common tumor biology, similar body size, and greater genome homology between dogs and humans than mice and humans in cancer-gene families. [3,4] Many groups are conducting studies to compare tumor biology between canine and human at the molecular level to complement clinical and histopathologic findings made in both species. [5–12] A primary example involving a common canine-human driver mutation is the parallel development of toceranib for treatment of mast cell tumors (MCT) in dogs and sunitinib for treatment of Gastrointestinal stromal tumors (GIST) in humans. This was based on the discovery that mutations in *KIT* [13,14] result in its constitutive activation and is the cause of MCT in dogs and GIST in humans. Given this common biology, the drug company Sugen was able to develop two virtually identical molecules in parallel, sunitinib for use in humans [15] and toceranib for use in dogs. [16]

However, most cancer modeling experiments are performed in rodents. [17] While rodent models have been valuable for investigating cancer mechanisms, they are not always representative of humans. For example, widely used mouse strains are inbred, and so have limited genetic variation across animals. [18] Laboratory mice are also raised in a relatively uniform environment and fed a standardized diet. This uniformity of rodents used in cancer studies is unlike the heterogeneous population of human cancer patients. Xenograft models, in which cancer cells are transplanted from human to an immune compromised mouse, limit their ability to accurately recapitulate cancer in humans. [19,20,21] These limitations of rodent models may explain many of the failures of biomarker targets and therapeutics to translate to humans. [22]

Access to high-quality canine samples is critical to understanding the complexities of canine cancer and potential translation of canine findings to human cancer research. [23–25]

This complexity may involve many genes, each with multiple splice variants and/or potential oncogenic mutations, combined with significant expression changes relative to the normal state. These effects are difficult to find using single gene analysis. The average expression of a group of genes, [26,27] however, may be easier to detect. If such averages are considered, data complexity is reduced, allowing elucidation of cancer similarities and differences within and between canine and human species. These similarities and differences are often studied by conducting two-condition, disease versus control experiments. Human or animal genomic experiments are performed to find and characterize genes that exhibit significant expression differences between the two conditions. These genes are then proposed as putative biomarkers candidates. However, without simultaneous analyses of other diseases and species it is impossible to know if the discovered biomarkers are specific to the disease and species of interest. [28]

We developed a combined experimental-bioinformatic protocol to generate canine cancer transcriptomic data and analyze it in the context of existing human data. The protocol, shown in Fig 1, mitigates somewhat issues discussed above, i.e., use of canines as opposed to rodents, access to high quality samples from established biobanks, tracking average expression values of gene groups as opposed to individual genes, and analysis of multi-disease, multi-species data as opposed to disease versus control data from one species. The individual components of this protocol are not unique, but their combination is uncommon. Execution of this protocol can increase the likelihood of quantifying relationships between canine and human cancer and establishing cancer biomarker signatures common to both species.

We obtained treatment-naive patient canine sample sets (frozen tumor, and frozen normal, typically healthy tissue taken as close to the tumor site as possible, Fig 1A) for each of five tumor histological types (osteosarcoma, T-cell lymphoma, B-cell lymphoma, pulmonary carcinoma, and melanoma) from the Pfizer-CCOGC (Canine Comparative Oncology and Genomics Consortium) Biospecimen Repository. [29] We prepared 75 samples for RNA-Seq analysis (Fig 1A). Each sample was split into 3 replicates to make 3 cDNA libraries for sequencing analysis (Fig 1B). All told 225 libraries were sequenced. The RNA-Seq sequence reads for the 5 cancers were analyzed together and genes (Fig 1C) were clustered into co-expression (Fig 1D) modules. [30] A co-expression module corresponds to a group of genes that have a similar expression pattern across samples analyzed. Genes in a co-expression module are of interest because such genes are thought to be members of the same biological pathways or part of the same transcriptional regulatory program. We found that each cancer exhibited specific activation for particular (primary) modules (Fig 1E). The co-expression modules were then used to build models (Fig 1F) that classified the canine cancers with high sensitivity and specificity (Fig 1G). These models also successfully classified matched human data. Genes were selected from those modules most highly activated for each cancer and proposed as diagnostic biomarker candidates (Fig 1H).

Many of the genes in these modules have little evidence of being targeted with drugs for the cancers studied. These genes, therefore, are good candidates for future target and/or drug discovery efforts. Annotations associated with these genes [31,32] revealed some common, but mostly distinct biological mechanisms for each cancer (Fig 1I). Most of these mechanisms are known, but they provide validation for the co-expression modules, and biomarker genes selected therefrom.

The work presented herein supports the comparative and clinical translational relevance of canine-to-human cancer studies, and the biological mechanisms at play that allow cancers to be accurately classified. The demonstrated transferability of canine derived diagnostic models to the human case reinforces the idea that biomarkers used in the detection and treatment of cancers are shared between canines and humans.

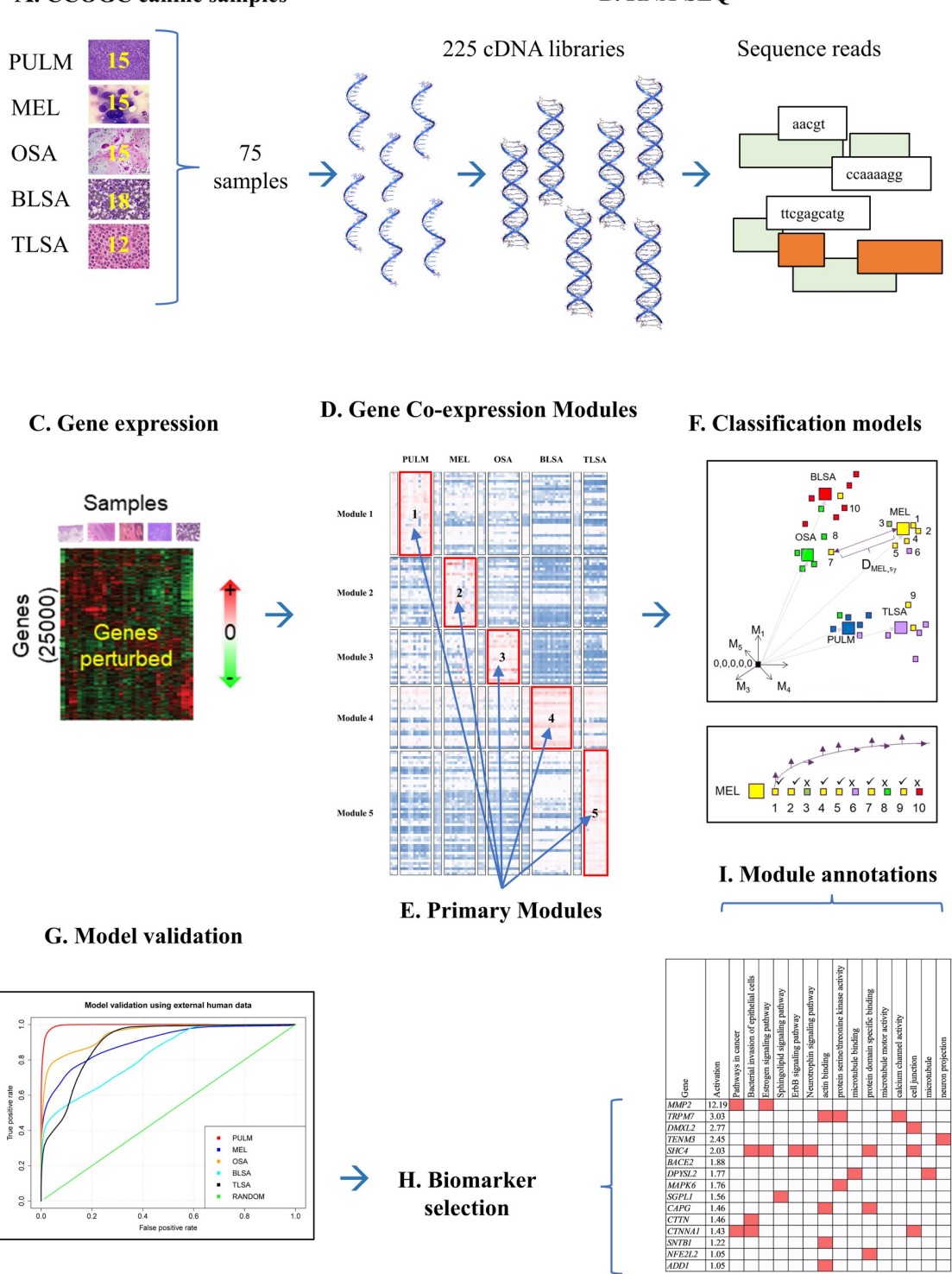

**Fig 1. Combined experimental-bioinformatics pipeline for multi-species, multi-disease, characterization, modeling, and biomarker hypothesis generation.**

## Results

### Five gene co-expression modules represent coordinated activation patterns of gene groups associated with 5 types of cancer

We obtained 75 tissue samples (S1 File, S1 and S2 Figs) associated with treatment naive canine patients from the Pfizer-CCOGC Biospecimen Repository. [29] These consisted of 12 tumor samples and 3 matched controls (taken near the vicinity of the tumor, if possible) for each of the 3 cancer types: pulmonary carcinoma (PULM), melanoma (MEL), and osteosarcoma (OSA). For B-cell lymphoma (BLSA) we collected 15 tumor samples and 3 matched controls and for T-cell lymphoma (TLSA) we collected 9 tumor samples and 3 matched controls. Histo-pathological evaluation by a veterinary pathologist indicated that tumor specimens on average contained 80% tumor cells and 20% stroma cells. The breed heterogeneity was high with 23 breeds represented in 60 dogs used for the study. One tumor biopsy for each sample was used and RNA was extracted and determined to be of high quality with all RIN values > 7.0. See S2 File for details. Each RNA sample from the single tumor biopsy was divided into 3 replicates. Each replicate was used to make 3 cDNA libraries for sequencing. [33,34] Each of the 3 librar-ies were loaded onto 3 different lanes on the flow cell that goes into the sequencer. 225 libraries in all were sequenced. See Methods section titled "**RNA-Seq Experiments**" for details.

After initial processing of the resulting sequence reads from RNA-Seq, we arrived at a 24,580 gene by 225 sample raw count matrix. See S3 File. We further processed the count matrix by (i) removing canine genes with low read counts, (ii) removing canine genes with low variance, (iii) removing canine genes with no associated human homologs, (iv) aggregat-ing sample replicates, and (v) performing a Z scaling transformation (see Eq (1), discussion in Methods). The final processed Z matrix (expression matrix, see S4 File) was composed of 4115 genes (rows) and 75 samples (columns). Relevant statistics associated with the Z matrix are: (i) range = -0.75 to 49.29, (ii) median = -0.18, and (iii) 0%, 25%, 50%, 75% and 100% quantiles = -0.75, -0.36, -0.18, 0.05, and 49.29, respectively. The distribution of Z scaled read counts, has a minimum value near zero, and is splayed in a positive direction for genes with increasing expression level. These Z scaled read counts, are essentially a measure of absolute activation with 0 being the reference and are read counts processed into a form and range amenable to module calculations. The positive splayed values of the Z matrix reflect the properties of the original read count matrix, from which it was derived (minimum value = 0, all counts positive).

Using an iterative clustering method, [30] we identified 5 gene co-expression modules asso-ciated with the Z matrix. The modules are given in S5 File. Fig 2 shows the expression patterns of each of these modules across the samples. The expression patterns of the modules across the cancer samples are distinct, typically exhibiting significant activation for one or two cancer types. Conversely each cancer exhibits a distinct expression pattern across modules, exhibiting significant activation for one or two modules.

### Each cancer type exhibits a distinct module activation profile corresponding to significant activation of one of the five modules

For each module, we define its activation as an average across all genes of the module and all samples associated with the cancer type or matched normal. The matched normal samples were taken from the same dog as the tumor samples and were typically healthy tissue taken near the tumor site. See S2 Fig for more information regarding the normal samples. The acti-vation values of the modules for cancer and normal samples are shown in Fig 3A and 3B. *P* val-ues associated with comparison of the module activation of each cancer to its activation in the

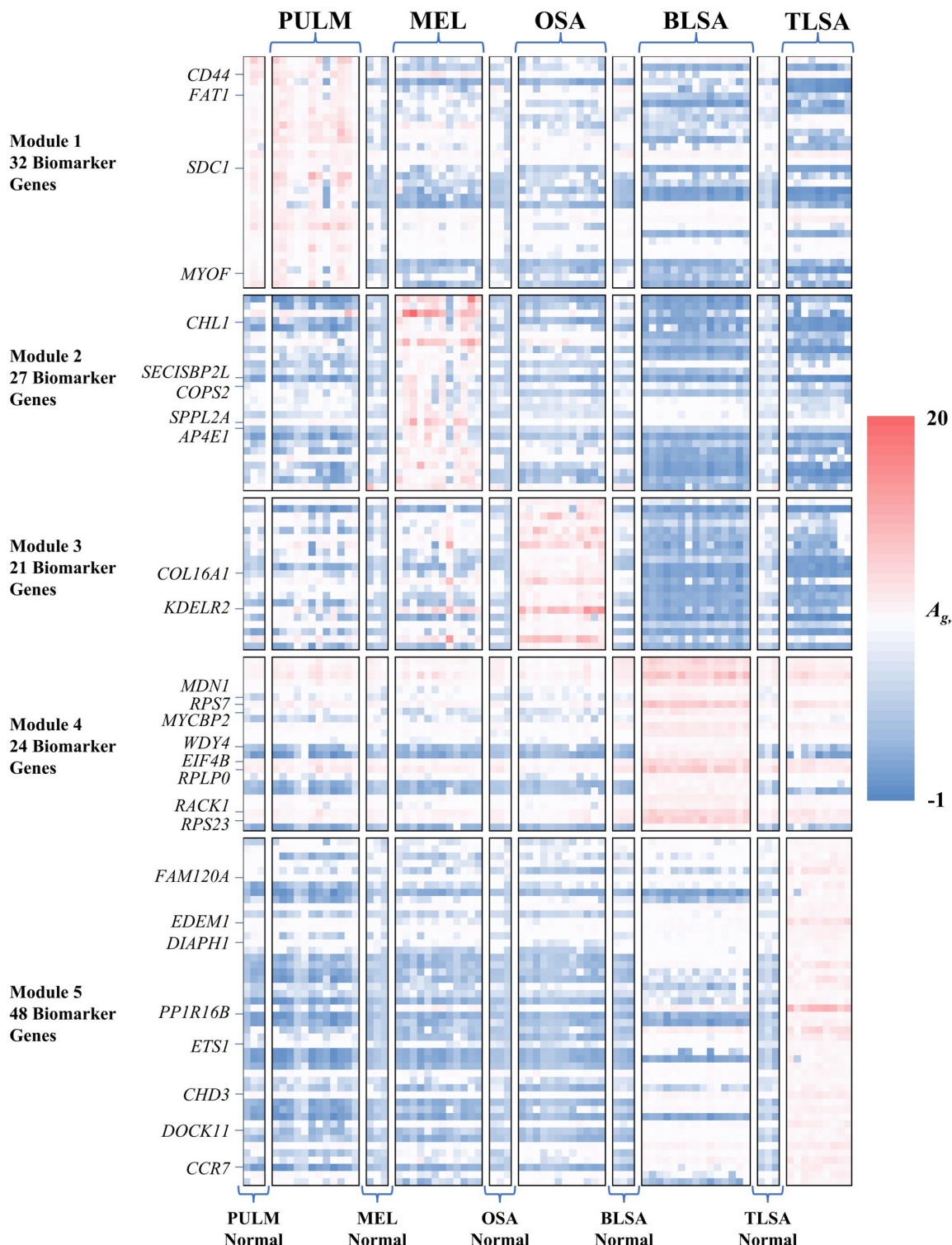

**Fig 2. Activation patterns of gene modules 1–5 across the cancer samples used in this study.** Rows represent individual module genes, columns represent samples, color coding represents activation ranging from low (blue) to high (red). Individual gene activations are Z-scaled read counts computed using Eq (2). Representative genes from each module are shown at left.

**A**

| Module | Module activations for each cancer type | | | | |
|---|---|---|---|---|---|
| | 1 | 2 | 3 | 4 | 5 |
| PULM | 0.82 | -0.15 | -0.17 | 0.27 | -0.14 |
| MEL | -0.08 | 0.73 | 0.02 | 0.19 | -0.12 |
| OSA | -0.10 | -0.13 | 0.89 | 0.12 | -0.10 |
| BLSA | -0.22 | -0.36 | -0.43 | 2.38 | 0.11 |
| TLSA | -0.26 | -0.38 | -0.45 | 0.71 | 1.38 |

**B**

| Module | Module activations for matched normals | | | | |
|---|---|---|---|---|---|
| | 1 | 2 | 3 | 4 | 5 |
| PULM | 0.60 | -0.05 | -0.12 | 0.17 | -0.13 |
| MEL | 0.06 | -0.02 | -0.10 | 0.33 | -0.11 |
| OSA | 0.05 | -0.04 | -0.09 | 0.33 | -0.09 |
| BLSA | 0.13 | -0.03 | -0.13 | 0.59 | -0.12 |
| TLSA | 0.07 | -0.04 | -0.09 | 0.46 | -0.08 |

**C**

| Module | $p$ * values, cancer versus matched normal | | | | |
|---|---|---|---|---|---|
| | 1 | 2 | 3 | 4 | 5 |
| PULM | 1.53E-03 | 2.32E-02 | 1.06E-01 | 6.38E-01 | 8.01E-01 |
| MEL | 9.56E-10 | 1.41E-21 | 3.76E-02 | 2.32E-02 | 2.67E-02 |
| OSA | 5.48E-10 | 2.47E-08 | 1.57E-27 | 4.03E-02 | 4.09E-02 |
| BLSA | 3.37E-19 | 5.77E-32 | 2.02E-30 | 7.69E-11 | 5.82E-04 |
| TLSA | 7.72E-29 | 2.81E-35 | 5.77E-32 | 5.65E-01 | 7.71E-46 |

* Wilcoxon rank sum test with Benjamini Hochberg correction

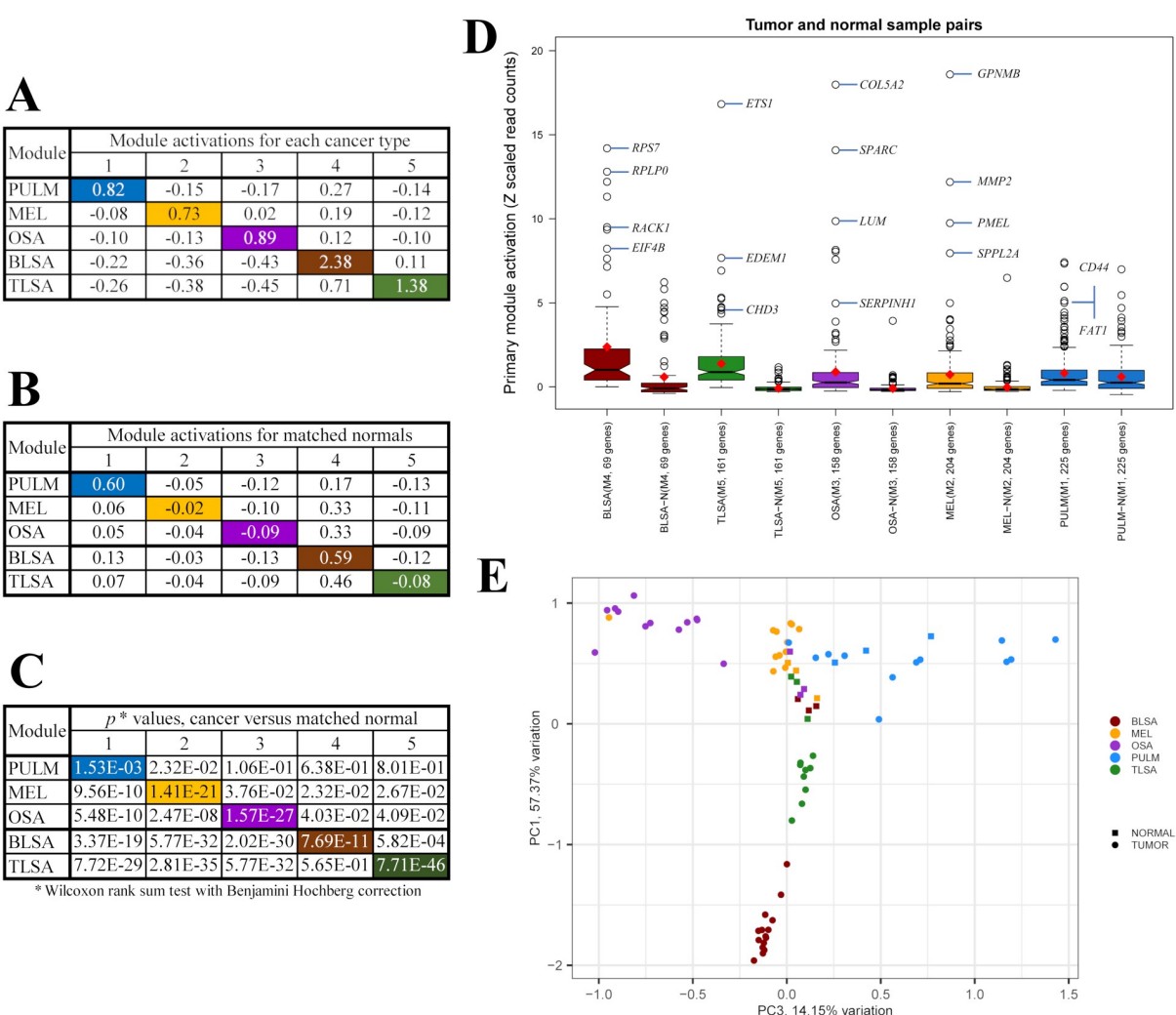

**Fig 3. Comparison of module activations between cancer and normal samples.** (A) Module activations for each of the 5 cancers studied. (B) Module activations of matched normal controls. (C) Module activation $p$ values for comparison of cancer sample activation with normal sample activation. Colored entries in (A)-(C) mark the values for the primary modules of each cancer type. Primary modules are defined as the highest expressed module with $p < 0.05$ for each cancer type. The primary modules are: module 1 (blue) for PULM, module 2 (orange) for MEL, module 3 (purple) for OSA, module 4 (brown) for BLSA, and module 5 (green) for TLSA. (D) Box plots showing distributions of primary module gene activations for each cancer and matched normal (Eq (3)). Open circles are oulier points. Some of the outlier points associated with the tumor samples are labeled with representative genes from each module. Red diamonds locate positions of the average activation across all genes within each module and all samples representing each cancer (Eq (5)). The center of each notch is the position of the median activation. (E) PCA plot showing positions of cancer samples (circles) and associated matched normals (squares). Each sample is represented by a set of 5 module activations, each activation compiled as an average across all genes of each module.

normal controls are shown in Fig 3C. In Fig 3A–3C, the colored boxes are associated with the primary module of each cancer type. The primary modules are defined as the most highly activated modules with statistically significant cancer sample activation relative to normal control activation. Module 1 (225 genes) is the primary module of PULM, module 2 (204 genes) is primary to MEL, module 3 (158 genes) is primary to OSA, module 4 (69 genes) is primary to BLSA and module 5 (161 genes) is primary to TLSA. Fig 3D shows notched box plots associated with the activation of primary module genes for each cancer and its matched normal. The non-overlap of notch positions for cancers versus normal for all cancer types indicates significant differences in gene activation between disease and normal states. The $p$ values shown in

Fig 3C confirm this. Fig 3E shows the positions of cancer (circles) and normal samples (squares) in PCA1, PCA3 space. Each sample is represented by a set of 5 module activations, with each activation compiled as an average across all genes of each module. The PCA plot shows excellent separation of the five sets of tumor samples. For BLSA, TLSA, MEL, and OSA, the normal samples are all similar, by their location near the center of the PCA plot. They are also different than the tumor samples which typically appear nearer the edges of the PCA plot. This implies that the normal samples were collected from similar tissues on the dogs but at locations distant from the tumors. In these cases, the normal tissue collected is most likely muscle or skin. The PULM normal samples, however, occupy the same region in the PCA plot as the PULM tumor samples. This implies that the PULM normal samples were collected from healthy lung tissue in the vicinity of the tumor.

### Nearest neighbor models based on module activation profiles successfully classified the canine samples used in this study

Because each cancer exhibits significant activation in different modules, we reasoned that the gene modules could be used to build accurate cancer classification models. We used a nearest neighbor approach to build the classification models. See the Methods section titled "**Model validation using internal canine data"** for details. When we tested this classification model for robustness using random subsample cross-validation on our internal data we obtained the set of ROC curves and associated metrics shown in Fig 4A and 4B. The average AUC and sensitivity (at a chosen specificity of ~0.7) across the ROC plots is 0.99 and 0.99 respectively. These results indicate that classification of tumor samples used in this study was robust using models built from gene co-expression modules.

### The nearest neighbor models successfully classified both canine and human transcriptomic data sets from external sources despite variability introduced by different technical methods, species, and sample types

Because we consider the average expression of gene groups which have lower variability than the expression values of single genes, we may be able to classify external canine data despite the experimental variability introduced by different technical methods. If human cancer biology is like canine, as we hypothesize, then the models may also classify external human data despite the variability introduced by different species.

Toward these ends, we tested our classification models for robustness on external data of the same cancer types for which the model was built. See the Methods section titled "**Model validation using external data"** for details. Using the external canine transcriptomics data (see S3 Fig, column titled "GEO series Dog") we obtained the set of ROC curves shown in Fig 4C, and associated metrics in Fig 4D. The average AUC and sensitivity at a chosen specificity of ~0.7 across the ROC plots is 0.97 and 0.99, respectively. Classification of external human transcriptomics data (see S3 Fig, column titled "GEO series Human") leads to the set of ROC curves shown in Fig 4E, and associated metrics in Fig 4F. The average AUC, sensitivity and specificity across the ROC plots is 0.95, 0.94, and 0.69, respectively. In a final test, we seeded the external human data (S3 Fig, column titled "GEO series Human") with negative data (human) from cancer types not used to construct the models (S4 Fig, data from the genomic data commons data portal). In this case the average AUC, sensitivity, and specificity across the ROC plots is 0.91, 0.91, and 0.70. These results indicate that our gene co-expression modules, derived from internal canine transcriptomic data, produce classification models, that also apply to external canine and human data for cancer types matched to those studied in this work (Fig 4A–4F). In addition, the models are selective for PULM, MEL, OSA, BLSA, and

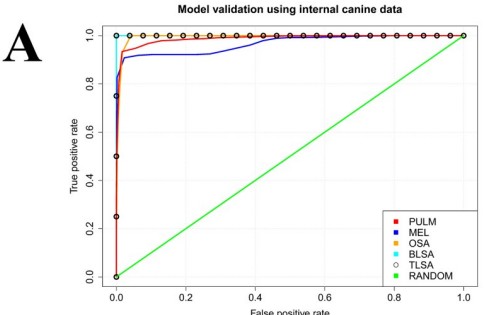

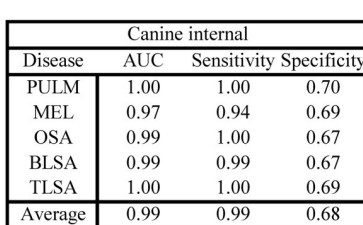

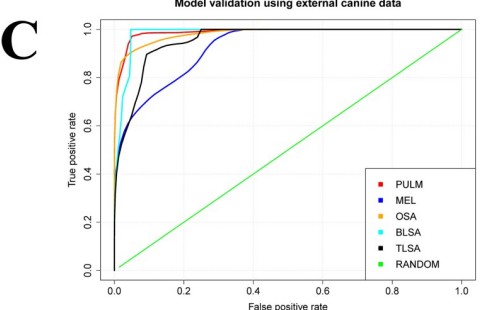

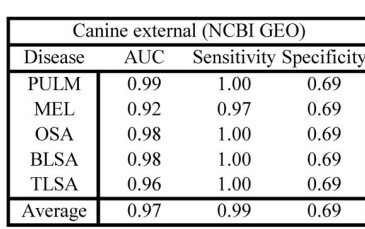

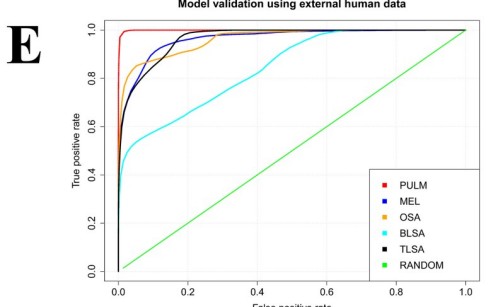

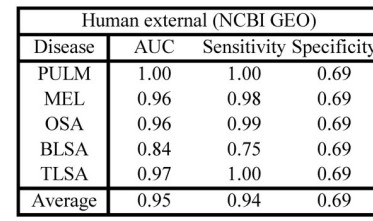

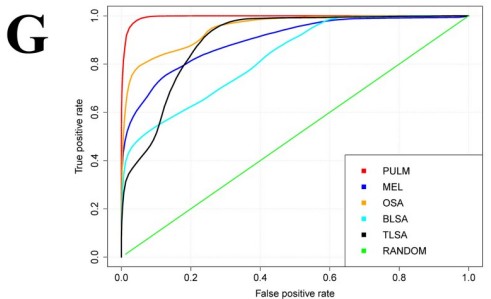

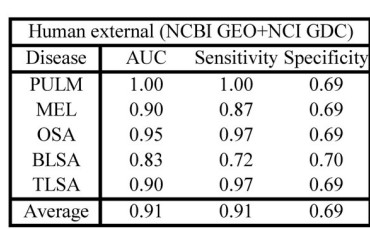

**Fig 4. Tests of cancer classification models.** (A)(B) ROC analysis of canine cancer models trained on half of the internal canine data and tested on the other half. (C)(D) ROC analysis of canine cancer models trained on all internal canine data and tested on external canine data of the same cancer type. (E)(F) ROC analysis of canine cancer models trained on all internal canine data and tested on external human data of the same cancer type. (G)(H) ROC analysis of canine cancer models trained on all internal canine data and tested on external human data of the same cancer types. External data sets for cancer types other than PULM, MEL, OSA, BLSA, and TLSA are added as background negative data.

TLSA even in the presence of data for cancer types not used to construct the models (Fig 4G and 4H).

Fig 4 exhibits the following trend in ROCS statistics from best to worse depending on the data sets used: Canine internal (AUC = 0.99), canine external (AUC = 0.97), human external (AUC = 0.95), human external augmented with cancer types other than MEL, PULM, OSA, BLSA, and TLSA (AUC = 0.91). The ROCs statistics appear to decay as the data sets used to test the classification models diverge from the original data set used to construct the models.

## Annotations reveal some common but mostly distinct and separate biologic themes associated with each cancer

We used the list of genes associated with each module to query the David bioinformatics resource [35] and found KEGG [30] and GO [32] annotations associated with each module. The annotation matrix is shown in Fig 5 for those annotations with $p$ values < 0.005. [36] 17 annotations are shared across multiple cancer types and 59 are cancer specific. The full annotation matrix ($p < 0.05$) for each cancer type is tabulated in S6 File (Tab 1). In this case 47 annotations are associated with multiple cancers and 279 are cancer specific. These statistics suggest that each gene co-expression module, to a significant extent, represents distinct annotation sets. Since each cancer studied in this work is closely related to specific primary modules, the annotations reveal some common but mostly distinct and separate biology associated with each cancer.

Fig 5 reveals some general themes. Annotations shared by 3 or 4 modules are rather general, e.g., "cytosol," "protein binding," and "ATP Binding." Annotations shared by two modules are more specific. For instance, modules 1 and 2, or 1 and 3 share annotations indicating biology related to cell-cell or cell ECM interactions, e.g., "cell-cell adherens junction," "cell adhesion," "extracellular matrix organization," and "cell matrix adhesion." Annotations shared by modules 2 and 3 focus on morphology, e.g., "angiogenesis," and "axon guidance." Annotations that focus on internal cell processes, e.g., "nucleoplasm," "nucleus," and "intracellular signal transduction" are shared by modules 4 and 5.

Annotations associated with one module only are the most specific of all, they are too numerous to include all here, so we highlight just a few. Module 1 annotations, associated with pulmonary carcinoma, focus on surfactants and vesicles, e.g., "surfactant homeostasis," "lysosome," and "positive regulation of extracellular exosome assembly." Module 2 annotations associated with melanoma focus on the cell cytoskeleton, e.g., "actin binding," "barbed-end actin filament capping," "actin filament binding, and "microtubule cytoskeleton organization." Module 3 annotations associated with osteosarcoma focus on bone components or processes, e.g., "collagen binding," "extracellular matrix," "skeletal system development," "ossification," and "calcium ion binding." Ribosomes and processes that occur therein are a focus for module 4 annotations associated with B-cell lymphoma, e.g., "ribosome," and "translation initiation." Module 5 annotations associated with T-cell lymphoma focus on T-cell activities, e.g., "T-cell receptor binding," "T-cell activation," "T-cell receptor signaling pathway," or second messenger systems, e.g., "Phosphatidylinositol signaling system," "peptidyl-tyrosine autophosphorylation," or epigenetic and transcriptional control, e.g., "histone-lysine N-methyltransferase activity," or "positive regulation of transcription-DNA templated."

## Biomarker hypotheses selected from the primary modules of each cancer bear significant evidence linking them to their associated cancer

A juxtaposition of annotations, genes, and modules, specific to the various cancer types is given in Figs 6–11. In these figures, for each cancer, and its associated modules we only

| Category | Annotation detail | Cancer type | | | | | # Modules | BH corrected p value |
|---|---|---|---|---|---|---|---|---|
| | | PULM | MEL | OSA | BLSA | TLSA | | |
| | | Module 1 | Module 2 | Module 3 | Module 4 | Module 5 | | |
| | | 225 genes | 204 genes | 158 genes | 69 genes | 161 genes | | |
| GO_CC | GO:0005829~cytosol | 1 | 1 | 0 | 1 | 1 | 4 | 1.64E-05 |
| GO_MF | GO:0005515~protein binding | 0 | 1 | 0 | 1 | 1 | 3 | 2.81E-07 |
| GO_MF | GO:0005524~ATP binding | 0 | 1 | 0 | 1 | 1 | 3 | 4.83E-03 |
| GO_CC | GO:0005913~cell-cell adherens junction | 1 | 1 | 0 | 0 | 0 | 2 | 3.16E-03 |
| GO_BP | GO:0098609~cell-cell adhesion | 1 | 1 | 0 | 0 | 0 | 2 | 4.26E-03 |
| GO_BP | GO:0030198~extracellular matrix organization | 1 | 0 | 1 | 0 | 0 | 2 | 1.04E-09 |
| GO_BP | GO:0007155~cell adhesion | 1 | 0 | 1 | 0 | 0 | 2 | 4.45E-07 |
| GO_CC | GO:0005789~endoplasmic reticulum membrane | 1 | 0 | 1 | 0 | 0 | 2 | 3.73E-04 |
| GO_CC | GO:0009986~cell surface | 1 | 0 | 1 | 0 | 0 | 2 | 3.28E-03 |
| GO_BP | GO:0007160~cell-matrix adhesion | 1 | 0 | 1 | 0 | 0 | 2 | 4.53E-03 |
| GO_BP | GO:0001525~angiogenesis | 0 | 1 | 1 | 0 | 0 | 2 | 3.32E-03 |
| KEGG | hsa04360:Axon guidance | 0 | 1 | 1 | 0 | 0 | 2 | 4.79E-03 |
| GO_CC | GO:0031234~extrinsic component of cytoplasmic side of plasma membrane | 0 | 1 | 0 | 0 | 1 | 2 | 1.60E-03 |
| GO_CC | GO:0005654~nucleoplasm | 0 | 0 | 0 | 1 | 1 | 2 | 4.73E-05 |
| GO_BP | GO:0035556~intracellular signal transduction | 0 | 0 | 0 | 1 | 1 | 2 | 4.64E-04 |
| GO_CC | GO:0005634~nucleus | 0 | 0 | 0 | 1 | 1 | 2 | 1.71E-03 |
| GO_BP | GO:0050853~B cell receptor signaling pathway | 0 | 0 | 0 | 1 | 1 | 2 | 4.83E-03 |
| GO_CC | GO:0005764~lysosome | 1 | 0 | 0 | 0 | 0 | 1 | 1.79E-04 |
| GO_CC | GO:0016021~integral component of membrane | 1 | 0 | 0 | 0 | 0 | 1 | 1.88E-04 |
| GO_BP | GO:0043129~surfactant homeostasis | 1 | 0 | 0 | 0 | 0 | 1 | 7.96E-04 |
| GO_BP | GO:1903553~positive regulation of extracellular exosome assembly | 1 | 0 | 0 | 0 | 0 | 1 | 3.75E-03 |
| KEGG | hsa05100:Bacterial invasion of epithelial cells | 0 | 1 | 0 | 0 | 0 | 1 | 6.03E-04 |
| GO_BP | GO:0010761~fibroblast migration | 0 | 1 | 0 | 0 | 0 | 1 | 1.24E-03 |
| KEGG | hsa04915:Estrogen signaling pathway | 0 | 1 | 0 | 0 | 0 | 1 | 1.71E-03 |
| GO_CC | GO:0005905~clathrin-coated pit | 0 | 1 | 0 | 0 | 0 | 1 | 1.72E-03 |
| GO_MF | GO:0003779~actin binding | 0 | 1 | 0 | 0 | 0 | 1 | 1.78E-03 |
| GO_BP | GO:0030516~regulation of axon extension | 0 | 1 | 0 | 0 | 0 | 1 | 2.18E-03 |
| GO_CC | GO:0030659~cytoplasmic vesicle membrane | 0 | 1 | 0 | 0 | 0 | 1 | 2.18E-03 |
| GO_BP | GO:0051016~barbed-end actin filament capping | 0 | 1 | 0 | 0 | 0 | 1 | 2.55E-03 |
| GO_MF | GO:0051015~actin filament binding | 0 | 1 | 0 | 0 | 0 | 1 | 3.30E-03 |
| GO_BP | GO:0010499~proteasomal ubiquitin-independent protein catabolic process | 0 | 1 | 0 | 0 | 0 | 1 | 3.38E-03 |
| GO_BP | GO:0000226~microtubule cytoskeleton organization | 0 | 1 | 0 | 0 | 0 | 1 | 4.83E-03 |
| GO_CC | GO:0005578~proteinaceous extracellular matrix | 0 | 0 | 1 | 0 | 0 | 1 | 1.38E-11 |
| GO_MF | GO:0005518~collagen binding | 0 | 0 | 1 | 0 | 0 | 1 | 7.94E-07 |
| GO_BP | GO:0030199~collagen fibril organization | 0 | 0 | 1 | 0 | 0 | 1 | 8.19E-07 |
| GO_CC | GO:0005788~endoplasmic reticulum lumen | 0 | 0 | 1 | 0 | 0 | 1 | 1.65E-06 |
| GO_CC | GO:0005581~collagen trimer | 0 | 0 | 1 | 0 | 0 | 1 | 1.64E-05 |
| GO_CC | GO:0005615~extracellular space | 0 | 0 | 1 | 0 | 0 | 1 | 2.24E-04 |
| GO_CC | GO:0031012~extracellular matrix | 0 | 0 | 1 | 0 | 0 | 1 | 4.07E-04 |
| KEGG | hsa04510:Focal adhesion | 0 | 0 | 1 | 0 | 0 | 1 | 9.66E-04 |
| GO_BP | GO:0001501~skeletal system development | 0 | 0 | 1 | 0 | 0 | 1 | 1.38E-03 |
| GO_MF | GO:0008201~heparin binding | 0 | 0 | 1 | 0 | 0 | 1 | 2.20E-03 |
| GO_CC | GO:0005576~extracellular region | 0 | 0 | 1 | 0 | 0 | 1 | 2.22E-03 |
| GO_BP | GO:0001503~ossification | 0 | 0 | 1 | 0 | 0 | 1 | 3.20E-03 |
| GO_CC | GO:0043235~receptor complex | 0 | 0 | 1 | 0 | 0 | 1 | 3.32E-03 |
| GO_MF | GO:0005509~calcium ion binding | 0 | 0 | 1 | 0 | 0 | 1 | 3.84E-03 |
| GO_BP | GO:0006364~rRNA processing | 0 | 0 | 0 | 1 | 0 | 1 | 1.38E-11 |
| GO_BP | GO:0006413~translational initiation | 0 | 0 | 0 | 1 | 0 | 1 | 3.56E-10 |
| GO_BP | GO:0006614~SRP-dependent cotranslational protein targeting to membrane | 0 | 0 | 0 | 1 | 0 | 1 | 1.19E-07 |
| GO_MF | GO:0044822~poly(A) RNA binding | 0 | 0 | 0 | 1 | 0 | 1 | 2.93E-07 |
| GO_BP | GO:0019083~viral transcription | 0 | 0 | 0 | 1 | 0 | 1 | 3.21E-07 |
| GO_BP | GO:0000184~nuclear-transcribed mRNA catabolic process | 0 | 0 | 0 | 1 | 0 | 1 | 4.45E-07 |
| KEGG | hsa03010:Ribosome | 0 | 0 | 0 | 1 | 0 | 1 | 7.16E-07 |
| GO_MF | GO:0003735~structural constituent of ribosome | 0 | 0 | 0 | 1 | 0 | 1 | 1.87E-06 |
| GO_BP | GO:0006412~translation | 0 | 0 | 0 | 1 | 0 | 1 | 8.23E-06 |
| GO_CC | GO:0022627~cytosolic small ribosomal subunit | 0 | 0 | 0 | 1 | 0 | 1 | 1.18E-05 |
| GO_CC | GO:0005840~ribosome | 0 | 0 | 0 | 1 | 0 | 1 | 2.76E-05 |
| GO_CC | GO:0015935~small ribosomal subunit | 0 | 0 | 0 | 1 | 0 | 1 | 3.30E-05 |
| GO_MF | GO:0003723~RNA binding | 0 | 0 | 0 | 1 | 0 | 1 | 2.52E-04 |
| GO_CC | GO:0005730~nucleolus | 0 | 0 | 0 | 1 | 0 | 1 | 1.32E-03 |
| KEGG | hsa04662:B cell receptor signaling pathway | 0 | 0 | 0 | 1 | 0 | 1 | 2.55E-03 |
| GO_BP | GO:0050732~negative regulation of peptidyl-tyrosine phosphorylation | 0 | 0 | 0 | 1 | 0 | 1 | 3.45E-03 |
| GO_BP | GO:0050852~T cell receptor signaling pathway | 0 | 0 | 0 | 0 | 1 | 1 | 2.72E-06 |
| KEGG | hsa04660:T cell receptor signaling pathway | 0 | 0 | 0 | 0 | 1 | 1 | 2.75E-06 |
| KEGG | hsa04070:Phosphatidylinositol signaling system | 0 | 0 | 0 | 0 | 1 | 1 | 1.65E-04 |
| GO_MF | GO:0018024~histone-lysine N-methyltransferase activity | 0 | 0 | 0 | 0 | 1 | 1 | 2.44E-04 |
| GO_MF | GO:0004715~non-membrane spanning protein tyrosine kinase activity | 0 | 0 | 0 | 0 | 1 | 1 | 4.64E-04 |
| GO_BP | GO:0030168~platelet activation | 0 | 0 | 0 | 0 | 1 | 1 | 5.64E-04 |
| GO_BP | GO:0045893~positive regulation of transcription, DNA-templated | 0 | 0 | 0 | 0 | 1 | 1 | 1.24E-03 |
| GO_BP | GO:0038083~peptidyl-tyrosine autophosphorylation | 0 | 0 | 0 | 0 | 1 | 1 | 2.22E-03 |
| KEGG | hsa04650:Natural killer cell mediated cytotoxicity | 0 | 0 | 0 | 0 | 1 | 1 | 2.41E-03 |
| KEGG | hsa05166:HTLV-I infection | 0 | 0 | 0 | 0 | 1 | 1 | 2.86E-03 |
| GO_BP | GO:0014066~regulation of phosphatidylinositol 3-kinase signaling | 0 | 0 | 0 | 0 | 1 | 1 | 3.02E-03 |
| GO_BP | GO:0050900~leukocyte migration | 0 | 0 | 0 | 0 | 1 | 1 | 3.30E-03 |
| GO_BP | GO:0042110~T cell activation | 0 | 0 | 0 | 0 | 1 | 1 | 3.36E-03 |
| GO_MF | GO:0042608~T cell receptor binding | 0 | 0 | 0 | 0 | 1 | 1 | 4.55E-03 |

**Fig 5. KEGG and GO annotations for modules and associated cancers.** Annotation matrix with rows enumerated by the different annotations and columns enumerated by cancer types and associated primary modules. Matrix elements are populated with 1 (and colored red) or 0 depending on whether the annotation is or is not associated with the module. The red color is included to highlight the block diagonal nature of the matrix. The column titled "# Modules" tallies the total number of modules that the annotations are associated with. The average BH corrected $p$ values for these associations are given in the last column of the Table. Only $p$ values < 0.005 are shown. $P$ values are shaded from red to

white, with the highest *p* values in red transitioning to the lowest *p* values in white. See S6 File (Tab 1) for the full set of annotations ($p < 0.05$).

included genes with activation > 1.5 (~95% confidence level for Z scores) as biomarker hypotheses and we only included annotations that are specific to each cancer. The biomarker hypotheses collected from Figs 6–11 are shown in Fig 12. See S7 File for expanded versions of Figs 6–11, that include all module genes and all annotations for each cancer type. See the Methods section titled "**Module annotations**" for a detailed description of these figures.

Examples of biomarker genes for pulmonary carcinoma (from Fig 6) that are associated with the cancer specific themes discussed in relation to Fig 5 above are: *CTSH*, *LPCAT1*, *NAPSA*, and *ADGRF5* which are associated with "surfactant homeostasis," genes *ASAH1*, *NAPSA*, *LRRK2*, and *CTSH* which are associated with "Lysosome," and genes *SDC4* and *SDC1* which are associated with "positive regulation of extracellular exosome assembly." For all module 1 biomarkers the average number of CTD (Comparative Toxicogenomics Database) [37] references relating each gene to pulmonary carcinoma = 243 (Fig 6 column titled "CTD reference count") and the average CTD inference score per gene = 54 (Fig 6 column titled "CTD inference score"). See the Methods section titled "**Biomarker link to literature and target novelty**" for discussion of CTD evidence. The CTD evidence indicates a sizable amount of literature implicating these module 1 biomarker genes to pulmonary carcinoma mechanism.

For melanoma specific themes, Fig 7 shows that *TRPM7* is associated with "actin binding," *DPYSL2* is associated with "microtubule binding" and "regulation of axon extension", and

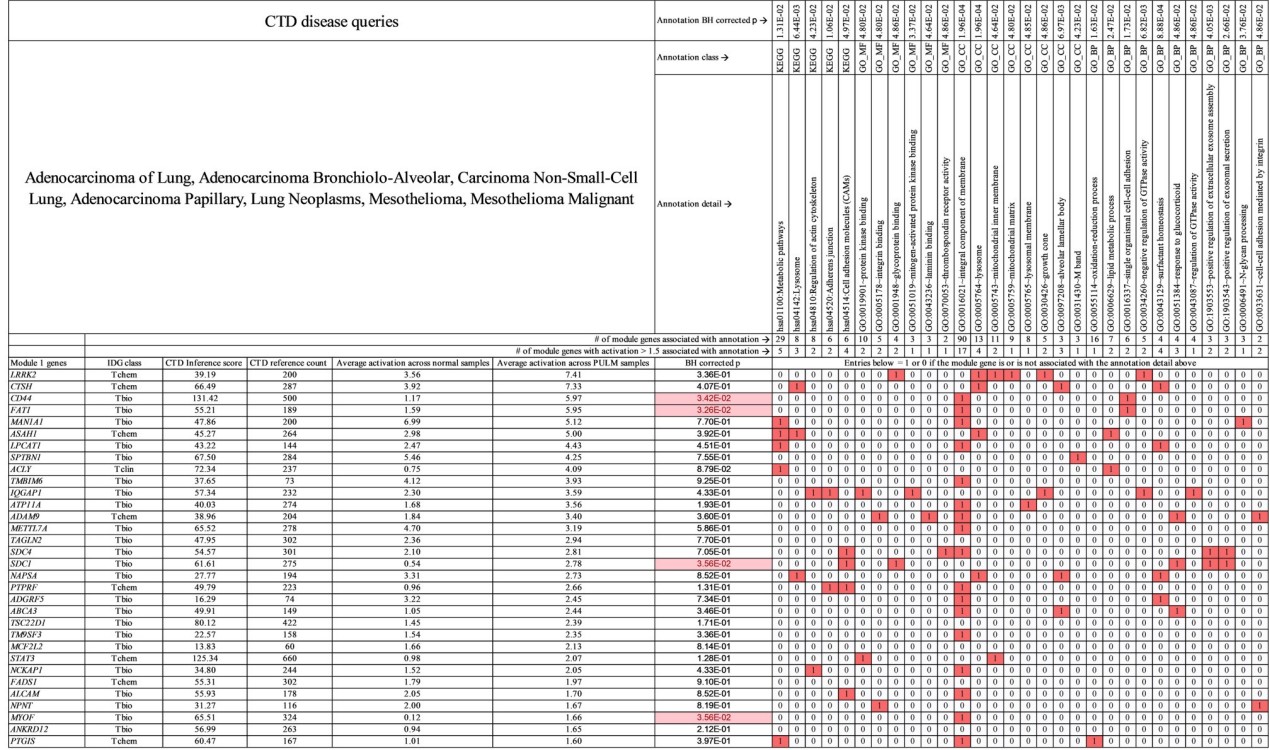

**Fig 6. Primary module 1 genes for pulmonary carcinoma.** Genes (activation > 1.5) and annotations shown are specific to pulmonary carcinoma in this study. In the table, red boxed 1 values indicate that the gene at far left in column 1 is associated with the annotation at top. White boxed 0 values indicate that the gene (at left) is not associated with the annotation (at top). Pink boxes under BH corrected *p* value indicate genes with statistically significant activation difference between tumor and normal samples. See S7 File which contains all module 1 genes for pulmonary carcinoma.

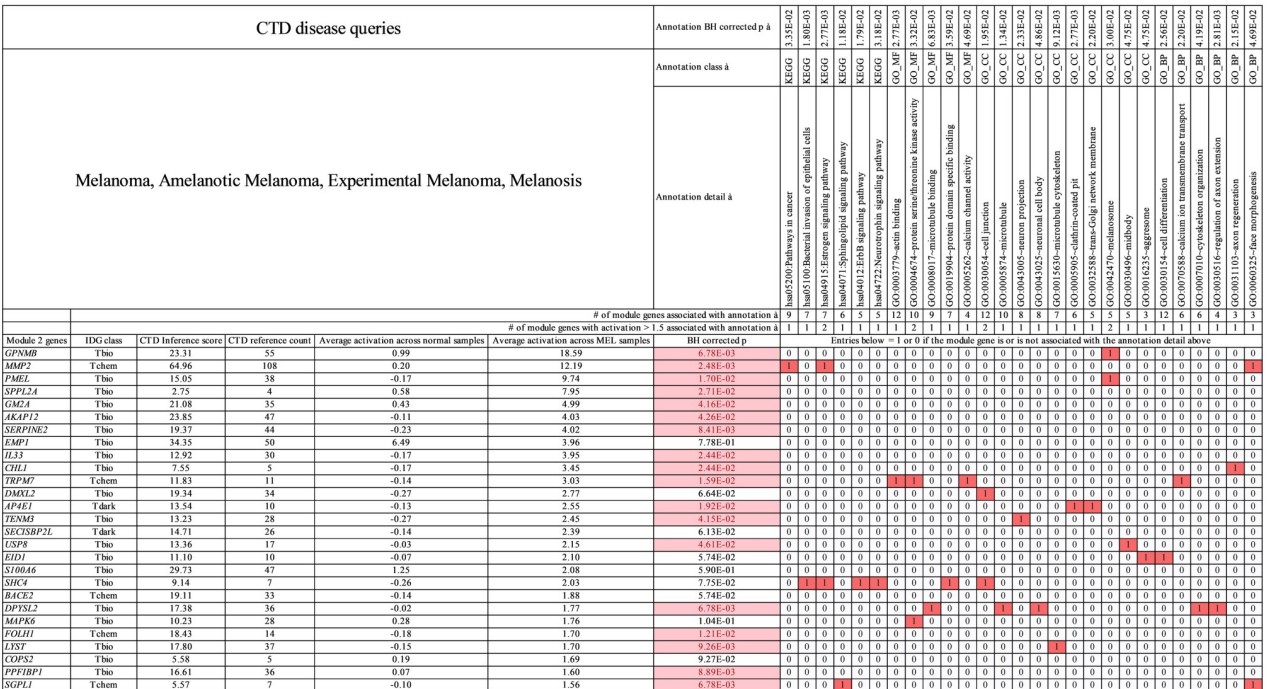

**Fig 7. Primary module 2 genes for melanoma.** Genes (activation > 1.5) and annotations shown are specific to melanoma in this study. In the table, red boxed 1 values indicate that the gene at far left in column 1 is associated with the annotation at top. White boxed 0 values indicate that the gene (at left) is not associated with the annotation (at top). Pink boxes under BH corrected *p* value indicate genes with statistically significant activation difference between tumor and normal samples. See S7 File which contains all module 2 genes for melanoma.

**Fig 8. Primary module 3 genes for osteosarcoma.** Genes (activation > 1.5) and annotations shown are specific to osteosarcoma in this study. In the table, red boxed 1 values indicate that the gene at far left in column 1 is associated with the annotation at top. White boxed 0 values indicate that the gene (at left) is not associated with the annotation (at top). Pink boxes under BH corrected *p* value indicate genes with statistically significant activation difference between tumor and normal samples. See S7 File which contains all module 3 genes for osteosarcoma.

**Fig 9. Primary module 4 genes for B-cell lymphoma.** Genes (activation > 1.5) and annotations shown are specific to B-cell lymphoma in this study. In the table, red boxed 1 values indicate that the gene at far left in column 1 is associated with the annotation at top. White boxed 0 values indicate that the gene (at left) is not associated with the annotation (at top). Pink boxes under BH corrected *p* value indicate genes with statistically significant activation difference between tumor and normal samples. See S7 File which contains all module 4 genes for B-cell lymphoma.

*LYST* is associated with "microtubule cytoskeleton." For all module 2 biomarkers the average number of CTD references relating each gene to melanoma = 30 and the average CTD inference score per gene = 17. These data indicate a smaller yet significant amount of literature linking module 2 biomarker genes to melanoma mechanism.

In the case of osteosarcoma specific themes, Fig 8 shows that *SPARK, LUM, SERPINH1, MRC2*, and *PCOLCE* are associated with "collagen binding," *COL5A2, LUM, COL6A3, COL12A1, TNC*, and *PCOLCE* are associated with "extracellular matrix," *COL5A2, COL12A1, CDH11*, and *ALPL* are associated with skeletal system development, *COL5A2, SPARC*, and *CDH11* are associated with "ossification," and *SPARC, CALU, CDH11*, and *FKBP10* are associated with "calcium ion binding." The module genes have average CTD number of references = 71 associating them with osteosarcoma and an average CTD inference score = 17. The CTD evidence linking module 3 genes to osteosarcoma is significant and greater than for melanoma, but less than that for pulmonary carcinoma.

For B-cell lymphoma themes, Fig 9 shows that *RPS7, RPLP0, RPS23, RPS12, RPL10A, RPS5, RPS16, RPL35*, and *RPS15* are associated with "ribosome," and *RPS7, RPLP0, RPS23, RPS12, E1F4B, RPL10A, RPS5, RPS16, RPL35, RPS15*, and *EIF3K* are associated with "Translational Initiation." The Module 4 genes have average CTD number of references = 6 associating them to B-cell lymphoma and average inference score = 7. These data indicate a smaller amount of literature linking Module 4 biomarker genes to B-cell lymphoma than for osteosarcoma, melanoma, and pulmonary carcinoma.

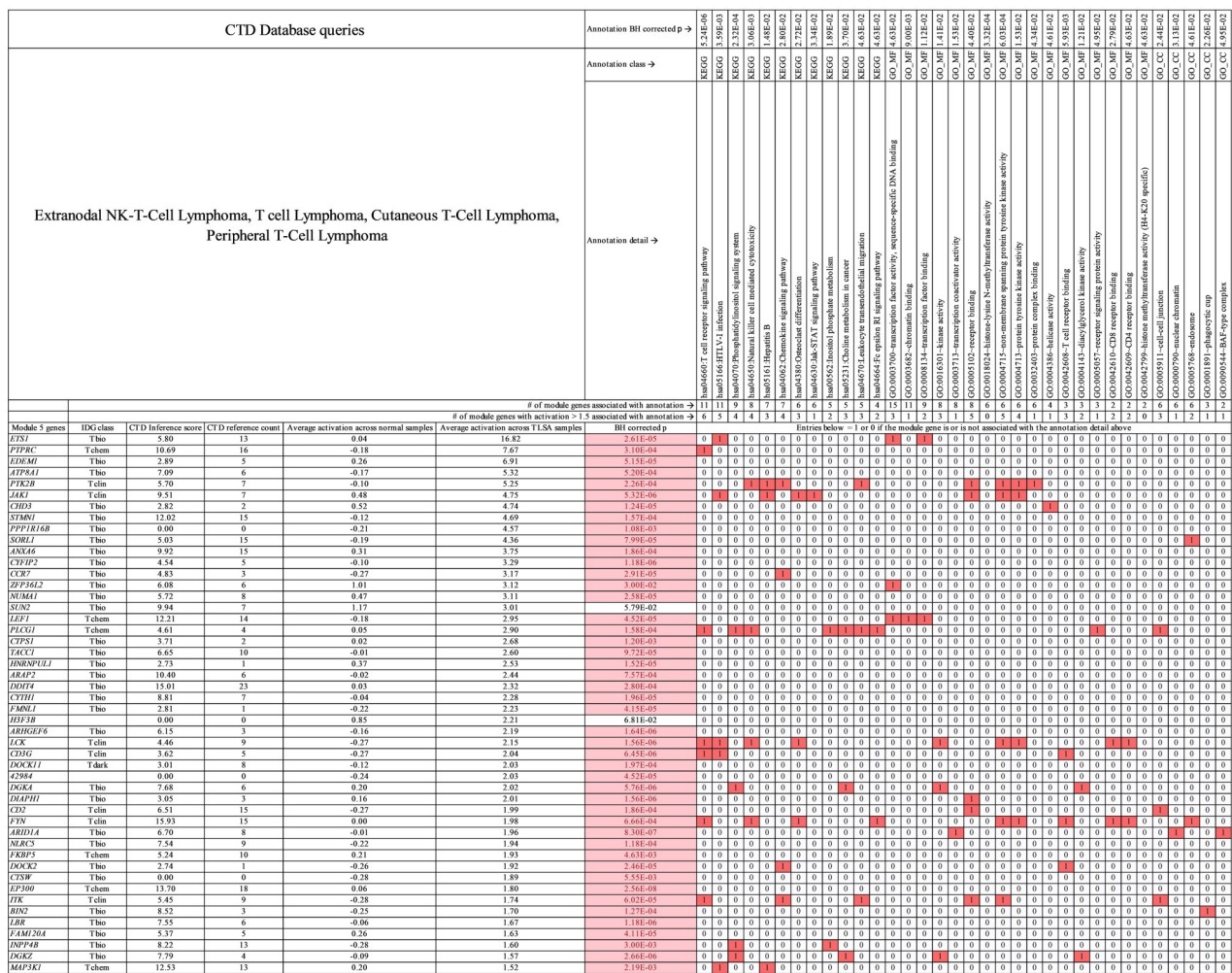

**Fig 10. Primary module 5 genes for T-cell lymphoma.** Genes (activation > 1.5) and annotations (KEGG, GO MF, and GO CC) shown are specific to T-cell lymphoma in this study. In the table, red boxed 1 values indicate that the gene at far left in column 1 is associated with the annotation at top. White boxed 0 values indicate that the gene (at left) is not associated with the annotation (at top). Pink boxes under BH corrected *p* value indicate genes with statistically significant activation difference between tumor and normal samples. See S7 File which contains all module 5 genes for T-cell lymphoma.

For T-cell lymphoma themes (Figs 10 and 11), genes *DOCK2*, *FYN*, and *CD3G* are associated with "T-cell receptor binding," *CD3G*, *CD2*, *FYN*, and *ITK* are associated with "T-cell activation," *PLCG1*, *DGKA*, *INPP4B*, and *DGKZ* are associated with "Phosphatidylinositol signaling system," *PTK2B*, *JAK1*, *LCK*, *FYN*, and *ITK* are associated with "peptidyl-tyrosine autophosphorylation," and *ETS1*, *LEF1*, and *ARID1A* are associated with "positive regulation of transcription-DNA templated." The module 5 genes have average CTD number of references = 8 and average inference score of 6 linking them to T-cell lymphoma. The CTD evidence linking module 5 genes to T-cell lymphoma is small like it is for module 4 gene linkage to B-cell lymphoma.

Further evidence linking module genes to their associated cancers was found by comparing module genes to cancer genes compiled in the COSMIC [38] (genes known to have mutations associated with specific cancers) and OncoKB [39] (known oncogenes) databases. In Fig 12 genes found in COSMIC are colored pink and those genes with bold surrounding boxes are found in OncoKB. For pulmonary adenocarcinoma, all module 1 genes are present in COSMIC and have documented mutations associated with the cancer. Four of the genes, *LRRK2*, *FAT1*,

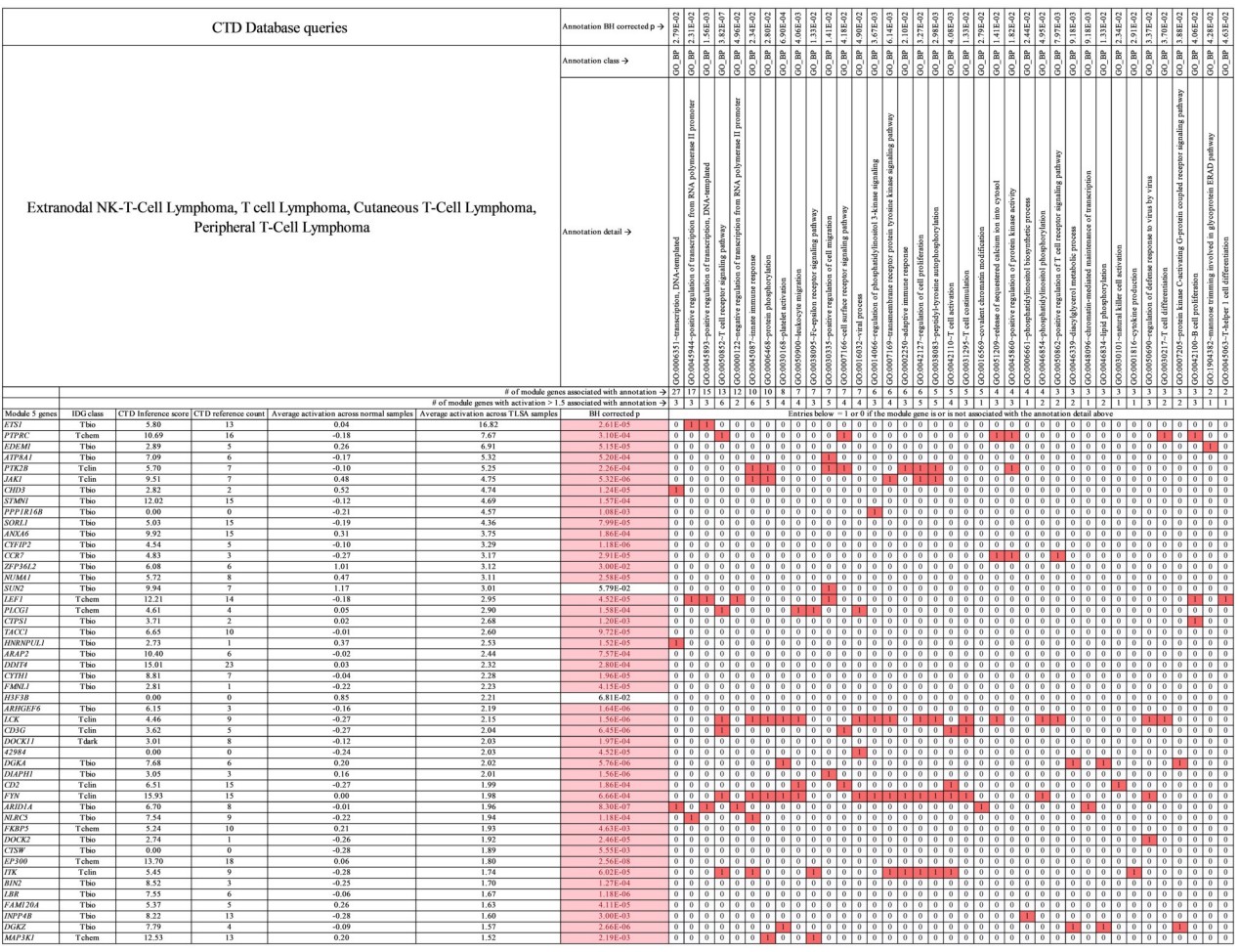

**Fig 11. Primary module 5 genes for T-cell lymphoma.** Genes (activation > 1.5) and annotations (GO_BP) shown are specific to T-cell lymphoma in this study. In the table, red boxed 1 values indicate that the gene at far left in column 1 is associated with the annotation at top. White boxed 0 values indicate that the gene (at left) is not associated with the annotation (at top). Pink boxes under BH corrected *p* value indicate genes with statistically significant activation difference between tumor and normal samples. See S7 File which contains all module 5 genes for T-cell lymphoma.

*SDC4*, and *STAT3* are present in OncoKB and are known oncogenes. For melanoma, all module two genes are present in COSMIC, and two of these genes, *USP8* and *PPFIBP1* are known oncogenes present in OncoKB. In the case of module 3 genes associated with osteosarcoma, eight are found in COSMIC (*LUM*, *COL6A3*, *TNC*, *CALU*, *COL16A1*, *OMD*, *ITGB5*, and *DPSYL3*), and two (*CDH11* and *OMD*) are oncogenes found in OncoKB. For B-cell lymphoma only three genes are found in cosmic (*CD22*, *MDN1*, and *PlCG2*) and three genes (*CD22*, *PTPN6*, *PLCG2*) are oncogenes. For T-cell lymphoma six genes are found in cosmic (*JAK1*, *PLCG1*, *FYN*, *ARID1A*, *EP300*, and *MAP3K1*), and 13 are known oncogenes. These 13 genes are *ETS1*, *PTPRC*, *JAK1*, *NUMA1*, *LEF1*, *PLCG1*, *LCK*, *FYN*, *ARID1A*, *EP300*, *ITK*, *INPP4B*, and *MAP3K1*.

## Many biomarkers selected from the primary modules of each cancer are underexplored as drug targets

Using a combination of Pharos IDG target class, [40] CTD inference score, and CTD number of references (columns 2, 3, and 4 of Figs 6–11) [41] we prioritized biomarkers in terms of

| KEY | Example |
|---|---|
| In COSMIC | GPNMB |
| In OncoKB | PTPN6 |
| Understudied drug target, IDG class "Tbio" | KDELR2* |
| Understudied drug target, IDG class "Tdark" | DOCK11** |

| Biomarker hypotheses from this work | | | | |
|---|---|---|---|---|
| Module 1 | Module 2 | Module 3 | Module 4 | Module 5 |
| PULM | MEL | OSA | BLSA | TLSA |
| LRRK2 | GPNMB | COL5A2 | RPL8 | ETS1 |
| CTSH | MMP2 | SPARC | RPS7* | PTPRC |
| CD44* | PMEL | LUM | RPLP0* | EDEM1* |
| FAT1* | SPPL2A* | COL6A3 | RPS23* | ATP8A1 |
| MAN1A1 | GM2A | COL12A1 | RACK1 | PTK2B |
| ASAH1 | AKAP12 | TNC | RPS12* | JAK1 |
| LPCAT1 | SERPINE2 | SERPINH1 | EIF4B | CHD3* |
| SPTBN1 | EMP1 | CALU | RPL10A* | STMN1 |
| ACLY | IL33 | COL16A1* | RPS5* | PPP1R16B* |
| TMB1M6 | CHL1* | CDH11 | RPS16* | SORL1 |
| IQGAP1 | TRPM7 | ALPL | NAP1L1 | ANXA6 |
| ATP11A | DMXL2 | MRC2 | MYCBP2* | CYFIP2* |
| ADAM9 | AP4E1** | PLOD2 | RPL35* | CCR7* |
| METTL7A | TENM3 | PCOLCE | CD22* | ZFP36L2 |
| TAGLN2 | SECISBP2L** | CALD1 | MDN1** | NUMA1 |
| SDC4 | USP8 | OLFML3 | RPS15* | SUN2 |
| SDC1* | EID1 | OMD | IMPDH2 | LEF1 |
| NAPSA | S100A6 | FKBP10 | WDFY4* | PLCG1 |
| PTPRF | SHC4 | ITGB5 | PTPN6 | CTPS1* |
| ADGRF5 | BACE2 | KDELR2* | PLCG2 | TACC1 |
| ABCA3 | DPYSL2 | DPYSL3 | SAMSN1 | HNRNPUL1* |
| TSC22D1 | MAPK6 | | USP34 | ARAP2 |
| TM9SF3 | FOLH1 | | FBL | DDIT4 |
| MCF2L2 | LYST | | EIF3K | CYTH1 |
| STAT3 | COPS2* | | | FMNL1* |
| NCKAP1 | PPFIBP1 | | | H3F3B* |
| FADS1 | SGPL1 | | | ARHGEF6* |
| ALCAM | | | | LCK |
| NPNT | | | | CD3G |
| MYOF* | | | | DOCK11** |
| ANKRD12 | | | | 42984 |
| PTGIS | | | | DGKA |
| | | | | DIAPH1* |
| | | | | CD2 |
| | | | | FYN |
| | | | | ARID1A |
| | | | | NLRC5 |
| | | | | FKBP5 |
| | | | | DOCK2* |
| | | | | CTSW* |
| | | | | EP300 |
| | | | | ITK |
| | | | | BIN2* |
| | | | | LBR |
| | | | | FAM120A* |
| | | | | INPP4B |
| | | | | DGKZ* |
| | | | | MAP3K1 |

**Fig 12. Biomarker candidates selected from Figs 6–11.** Genes colored pink are found in COSMIC, those with bold surrounding boxes are found in OncoKB, those with a single asterisk are IDG class "Tbio", and those with a double asterisk are IDG class "Tdark". Presence in COSMIC means that there is documented evidence of the gene being mutated for the specific cancer, presence in OncoKB means that the gene is a known oncogene, an IDG target class of "Tbio" or "Tdark" means that the gene is understudied as a drug target, i.e., there are no drugs or small molecules active against these targets.

their "target novelty" defined as targets with few chemistries (approved drugs and other) that have known activities against them. See the Methods section "**Biomarker link to literature and target novelty"** for details regarding this prioritization. In Fig 12 high priority targets are marked with an asterisk. These targets have the IDG classification "Tbio" which means that they do not have known drug or small molecule activities, but they do have associated gene ontology terms or a confirmed OMIM [42] phenotype. These targets also have a small CTD inference score and few CTD references linking them to their associated cancer (see columns 3 and 4, Figs 6–11). Highest priority targets, in terms of novelty, are marked with a double asterisk. These targets have the IDG classification "Tdark" which means that they are essentially unstudied.

In the case of pulmonary carcinoma (PULM), high priority targets are *CD44*, *FAT1*, *SDC1*, and *MYOF*. These targets all share the annotations, "extracellular exosome" and "plasma membrane". In the case of melanoma (MEL), high novelty targets are *SPPL2A*, *CHL1*, *AP4E1*, *SECISBP2L*, and *COPS2*. These targets are associated with a variety of annotations which include "extracellular exosome", "clathrin-coated pit", "trans-Golgi network membrane", "signal transduction", and "axon regeneration." For osteosarcoma, high novelty targets are *COL16A1* and *KDELR2*, and are associated with annotations that include "extracellular exosome", "endoplasmic reticulum lumen", "Golgi membrane", "cell adhesion", "extracellular matrix organization", and "integrin mediated signaling pathway".

There are many more high novelty targets associated with B and T-cell lymphoma, than for the other cancers. For B-cell lymphoma, high novelty targets are *RPS7*, *RPLP0*, *RPS23*, *RPS12*, *RPL10A*, *RPS5*, *RPS16*, *MYCPB2*, *RPL35*, *CD22*, *MDN1*, *RPS15*, and *WDY4*. Most of these targets are components of the ribosome and are involved in the processes that occur therein. For T-cell lymphoma, high novelty targets are *EDEM1*, *CHD3*, *PPP1R16B*, *CYFIP2*, *CCR7*, *CTPS1*, *HNRNPUL1*, *FMNL1*, *H3F3B*, *ARHGEF6*, *DOCK11*, *DIAPH1*, *DOCK2*, *CTSW*, *BIN2*, *FAM120A*, and *DGKZ*. Many of these targets are associated with cell signaling and second messenger systems and are associated with annotations that include "ATP binding", "GTPase activator activity", "nucleus", "intracellular", and "transcription, DNA-templated". For more details on target novelty please see the Methods section titled "**Biomarker link to literature and target novelty"**.

We compared the dog breeds used in our study, with the dog breeds contained in the American Kennel Club Database (https://www.akc.org/). The distribution of dog breeds represented in the tumor samples we chose for analysis roughly corresponds to the distribution of dogs owned in the USA (according to the American Kennel Club data). Our top breeds, by sample count were Golden Retriever, Mixed, Labrador Retriever, Boxer and Doberman. These are also the most popular breeds to be registered by the American Kennel Club. Since we are studying dog breeds commonly owned by people in the USA, the biomarker results presented here are likely to be widely applicable. See S1 File (last tab titled "AKC_comparo") for details.

## Discussion

Most cancer modeling experiments are performed on rodents bearing experimentally induced disease. [17] The shortfalls of these models may explain many of the failures of therapeutic and biomarker hypotheses to translate to humans in the clinic. [22] Pet dogs spontaneously develop malignancies that have common morphologic, histologic, and biologic characteristics with human cancers. [1,2] This has highlighted the potential for inclusion of such animals in biomarker and drug development studies as a complement to rodent models. Towards these ends we have now identified five gene co-expression modules from bioinformatic analysis of transcriptomic data for canine pulmonary carcinoma, melanoma, osteosarcoma, B- and T-cell

lymphoma. We showed that each of the canine cancers exhibits a unique module activation profile, represented by statistically significant expression of one or two of the modules (primary modules) specific to the cancers. We used these unique profiles to develop models that successfully classified the canine cancer data to high sensitivity and specificity. We showed that the models were also successful in classifying human transcriptomic data for the same five cancer types, thereby demonstrating canine-human equivalence of the models. Annotations of the primary modules of each cancer yielded some common but mostly cancer-specific biology. We selected biomarker candidates from the highest expressing genes of the primary modules of each cancer. Many of the biomarker genes turn out to be underexplored as drug discovery targets and warrant further study.

Identification of gene modules that can be used to provide accurate classification of multiple cancer types and are applicable to multiple species is a difficult endeavor. The success of the current study depends on a combination of aspects that taken together are not commonly applied.

First, because of our access to the Pfizer-CCOGC Biospecimen Repository, [29] we were able to collect a large set of samples from naturally occurring tumors extracted from treatment naïve dogs. Sample availability such as this is uncommon in the preclinical research setting. [43] We collected 12 tumor samples and 3 matched controls for MEL, OSA, and PULM. We collected 9 tumor samples and 3 matched controls for TLSA and 15 tumor samples and 3 matched controls for BLSA. The samples, on average, contained 80% tumor cells and 20% stroma cells. The breed heterogeneity was high with 23 breeds represented in 60 dogs used for the study. However, neither the tumor/stroma ratios nor the breed heterogeneity were confounding factors in our experiments. Please see S8 and S9 Files for details.

From Fig 3E we surmised that the normal samples for BLSA, TLSA, MEL, and OSA were collected from a common tissue source, most probably muscle or skin at a site remote from the tumors. The PULM normal samples, on the other hand, were collected from healthy lung tissue, in the vicinity of the tumor. To validate these observations, we compiled a list of the top 200 expressing (read counts) genes for each normal sample, and then performed a tissue specific enrichment analysis using the Dougherty Web-based TSEA tool. [44] The tissue marker genes pulled for the various normal samples, without a doubt, identified the B and T cell lymphoma normal tissues as skin with an admixture of adipose tissue, the osteosarcoma and melanoma normal samples were mainly skin with an admixture of muscle tissue, and the pulmonary carcinoma normal samples were mainly lung with an admixture of muscle tissue. These results are consistent with the positions of the normal samples on the PCA plot of Fig 3E. See S2 Fig for details. The result of these observations means that for PULM, it may be possible to deconvolute tissue specific transcription from cancer specific transcription based on the $p$ values for comparison of tumor to normal gene expression. For the other tumor types, this deconvolution based on $p$ values is not possible because the normal samples come from a different tissue source than the tumors. We may, however, use the Z values in these cases to determine if the genes are significantly expressed relative to background by considering only Z values $\geq 1.5$ (~95% confidence level for Z scores). Using these genes, it then may be possible, to use their associated annotations and inclusion, or not, in COSMIC [38] and OncoKB [39] to decide if module genes are related to cancer and not simply tissue specific expression.

Second, rather than focusing on individual genes, we focused on gene co-expression modules. Expression values of these modules are computed as an average across module genes. Consideration of a few gene modules and their averaged expression simplifies the system considerably and is likely to be more consistent across cancer samples, than the expression of individual genes. See S8 and S9 Files for details regarding stability of module expression as it relates to tumor/stroma ratio and dog breed heterogeneity.

There are many publications that describe the use of gene groups and their averaged expression to characterize cancers. Jin et al. [45] conducted a gene co-expression model analysis of public available osteosarcoma canine and human data. They found three human modules, eight canine modules, and four consensus modules, containing significant numbers of genes common between the canine and human modules. We compared our module 3 osteosarcoma genes (Fig 8 and S7 File) with the four consensus modules of Jin et al. [45] and found that six of our biomarker candidates were also in Jin's consensus module 1, these genes are *SPARC*, *COL6A3*, *PCOLCE*, *CALD1*, *ITGB5*, and *DPYSL3*. We found that two of these genes, *COL6A3* and *ITGB5*, as well as *TNC* and *COL5A2* are significantly expressed relative to background in module 3 and are associated with the focal adhesion-PI3K-Akt signaling axis in canine osteosarcoma as seen in Fig 8 and S7 File. In addition, somatic mutations in COL6A3 and ITGB5 have been documented for osteosarcoma samples in COSMIC [38]. *COL6A3*, *COL5A2*, and *TNC* code for extracellular matrix proteins. *ITGB5* codes for a membrane bound integrin, important for attachment of the cell to ECM. *TNC* inhibits the interaction of some ECM proteins to integrins [46,47] thereby weakening the interaction between tumor cells and ECM. This increases tumor cell mobility leading to metastasis. Further along the focal adhesion-PI3K-Akt axis, *ITGB5* interacts with *PTK2* (not a gene found here but known to interact with *ITGB5*) a focal adhesion kinase (*FAK*). FAKS are phosphorylated in response to integrin engagement with the ECM. Activated *FAK* augments *PI3K* levels [48] which then catalyze the formation of *PIP3* [49]. *PIP3* recruits *AKT* and *PDK1* to the membrane where *PDK1* phosphorylates *AKT* [49]. Activated *AKT* leads to a variety of outcomes including cell proliferation and angiogenesis (via NOS proteins). *PTEN* a tumor suppressor downregulates *PIP3* thereby inhibiting downstream activities of AKT. Oncogenic mutations are noticeably absent from the focal adhesion end (*COL6A3*, *COL5A2*, *TNC*, *ITGB5*) of the PI3K-Akt axis. However, oncogenic mutations occur downstream of *ITGB5* and include loss of function mutations in *PTEN* [50] and gain of function mutations in *FAK* [51] or *PI3K* [50].

Previous work has shown that the PI3K-Akt pathway is dysregulated in canine osteosarcoma [52] and frequently hyperactivated in human osteosarcoma [53]. The fact that *COL6A3*, *COL5A2*, *TNC*, and *ITGB5* which are highly activated here and part of the focal adhesion-PI3K-Akt signaling axis implies that they are central to osteosarcoma and not simply tissue specific activation. The activation of PI3K-Akt in canine and human osteosarcomas demonstrated here and the overlap of our canine osteosarcoma biomarkers with the common canine-human modules of Zin et al., demonstrates potential transferability of our canine osteosarcoma biomarkers to humans.

In another example using gene groups and averaged expression to characterize cancers, Loeffler-Wirth et al. [54] applied self-organizing map (SOM) machine learning to B-cell lymphoma transcriptomic data on 873 human biopsy specimens to cluster genes into 13 spot modules. We compared our canine derived module 4 genes (Fig 9 and S7 File) to the human derived spot modules of Loeffler-Wirth and found that 7 of our module 4 genes were also in the Loeffler-Wirth spot modules. These genes are *IMPDH2*, *SAMSN1*, *PA2G4*, *DNAAF2*, *AHCY*, and *UPT20*. Two of these genes, *IMPDH2* and *SAMSN1* are highly expressed genes in our list of biomarker candidates for B-cell lymphoma. *SAMSN1*, typically highly expressed in lymphomas [55], is a splice variant of *HACS1* which promotes RAC1-dependent membrane ruffle formation. Membrane ruffling is an indicator of tumor cell motility and metastatic potential. [56] *IMPDH2* is involved in the synthesis of GTP which is an energy source for ribosome biogenesis and protein production. [57] In module 4 there are many significantly expressed genes that code for structural components of the ribosome, e.g., *RPS7*, *RPLP0*, *RPS23*, *RPS12*, *RPL10A*, *RPS5*, *RPS16*, *RPL35*, and *RPS15*. Upregulation of these components implies ribosome biogenesis and is consistent with upregulation of *IMPDH2*. In dividing cells

extreme up-regulation of *IMPDH2* can lead to cell proliferation and malignancy in B- and T-cell lymphoma as well as other cancers. [58] Mutations in *IMPDH2* leading to B-cell lymphoma have not been reported but constitutive activation of *MYC*, its controlling transcription factor have. [59] In B- cell lymphoma this constitutive activation can occur via translocation, commonly resulting in *MYC-IGH* (immunoglobulin heavy chain) fusions. Constitutive activation of *MYC* can lead to the same for *IMPDH2* which produces energy and proteins necessary for construction of proliferating cells.

Annotations showing the relationship of *SAMSN1* to membrane ruffling and *IMPDH2* to control by *MYC* which in B-cell lymphoma is constitutively activated by fusion to *IGH*, implies that up regulation of *SAMSN1* and *IMPDH2* found here is associated with B-cell lymphoma and not necessarily tissue specific transcription. In addition, the overlap of our canine derived B-cell lymphoma biomarker genes with the human derived spot modules of Loeffler-Wirth et al., implies potential transferability of our canine derived biomarkers to humans.

Jeffs et al. [60] performed DNA microarray analyses on 28 cell lines generated from human melanoma tumor samples. Using hierarchical clustering they were able to partition the cell lines into two groups and they identified a cluster of 106 genes exhibiting a different expression pattern for each cell line group. Group 1 cells showed decreased expression of 66 genes (sub-group 1) involved in neural crest and melanocyte development, differentiation, and pigmentation and upregulation of 40 genes (sub-group 2) related to angiogenesis, neurogenesis, immunomodulation, and ECM remodeling. Group 2 cells exhibited the opposite expression pattern across the 106 genes, i.e., up regulation of sub-group 1 genes and down regulation of sub-group 2 genes. Cell motility measurements showed that Group 1 cells exhibited a 23-fold higher capacity for migration than group 2 cell lines. We compared our module 2 genes for melanoma (Fig 7 and S7 File) with the 106 gene cluster of Jeffs et al. and found an overlap of eight genes. These genes along with their activation values Eq (3) for melanoma (MEL) are, *GPNMB* (MEL activation = 18.59), *UBL3* (MEL activation = 1.39), *AP1S2* (MEL activation = 1.28), *RAB38* (MEL activation = 0.91), *EDNRB* (MEL activation = 0.55), *JUN* (MEL activation = 0.34), and *C1orf21* (MEL activation = -0.13). Limiting consideration to genes with significant expression relative to background (Z > 1.5) we retain *GPNMB* only.

*GPNMB* with expression value of 18.59 is significantly expressed for melanoma in this study and is characteristic of Jeffs non-migratory sub-group 2 human cell lines showing up regulation of this gene. This implies that our melanoma samples are non-metastatic and is consistent with the fact that we acquired them from primary tumors of dogs. Jeff's studies were performed on human cell lines, while ours were performed on canine tissue. The consistency between our results and Jeff's indicates that the canine melanoma genes described here (in particular, GPNMB) may potentially be useful biomarkers targets in both canines and humans.

*GPNMB*, the most highly expressed gene in primary module 2 of melanoma, locates to the surface of melanoma cells. *GPNMB* is required for melanin production in melanocytes. Mutations in *GPNMB* are reported in COSMIC but it is not clear if they are causative events in melanoma. *GPNMB* is related to mutations in *BRAF* and *MEK* that do cause melanoma. Most melanomas contain mutant *BRAF* or *MEK* subtypes. Melanoma develops resistance to BRAF inhibitors (vemurafenib, dabrafenib) and *MEK* inhibitors (trametinib, cobimetinib). Resistance occurs via increased expression of *MITF* a transcription factor. The expression levels of *GPNMB*, a target gene of *MITF*, likewise increase. High expression of *GPNMB* promotes tumor migration and invasion by interacting with integrins to localize immune-suppressive and pro-angiogenic cells to the tumor [61] and renders melanoma resistant to *BRAF* and *MEK* inhibitors. Combination therapy, however, which includes a *BRAF* or *MEK* inhibitor, combined with glembatumumab vedotin (an antibody drug conjugate that targets *GPNMB*) inhibits *GPNMB* and effectively controls the growth of Melanoma. [62] The connection of *GPNMB*

expression to mutations in *BRAF* and *MEK* which are common to melanoma implies that the high expression of *GPNMB* found in this study is a manifestation of melanoma and not simply tissue specific transcription.

Third, success of the current study also depends on the fact that we analyzed transcriptomic data on multiple cancer types together. This allows for discovery of gene modules that are specific to one or another cancer type. Our analysis of canine transcriptomic data found 5 co-expression modules and identified modules 4, 2, 3, 1, and 5 as primary modules to BLSA, MEL, OSA, PULM, and TLSA respectively (Fig 3A). Such modules provide a good basis for classification models (Figs 3E and 4). This is not possible in typical two state studies, i.e., disease versus control. A recent example related to our work can be found in the work of Peng et al.[26] They analyzed 7 human cancers together, clustered genes into co-regulated sets and from these identified a 14 gene signature able to distinguish primary human cancer samples from normal controls. In addition, they found a lung cancer-specific gene signature, containing *SFTPA1* and *SFTPA2* genes, that was able to distinguish lung cancer from the other cancers. These *SFTPA* genes code for type A lung surfactant proteins. Our current results for canine pulmonary carcinoma are related to the human results of Peng et al. but focus on those genes involved in the synthesis and trafficking of lung surfactant proteins. The genes involved, all associated with the GO BP term "surfactant homeostasis", (Fig 6 and S7 File) are *CTSH* (PULM activation = 7.33, Normal activation = 3.92, $p$ = 4.07E-01), *LPCAT1* (PULM activation = 4.43, Normal activation = 2.47, $p$ = 4.51E-01), *NAPSA* (PULM activation = 2.73, Normal activation = 3.31, $p$ = 8.52E-01) and *ADGRF5* (PULM activation = 2.45, Normal activation = 3.22, $p$ = 7.34E-01). However, all $p$ values are > 0.05 for comparison of tumor to normal expression. Previously we identified the normal samples for pulmonary carcinoma as mainly lung tissue with an admixture of muscle. That being the case, the high $p$ values indicate that the activation of these surfactant genes is a tissue specific effect.

High expression genes (activation > 1.5) in module 1 that do show statistically significant tumor-normal expression differences (Fig 6 and S7 File) are: *CD44* (PULM activation = 5.97, normal activation = 1.17, $p$ = 3.42E-02), *FAT1* (PULM activation = 5.95, normal activation = 1.59, $p$ = 3.26E-02), *SDC1* (PULM activation = 2.78, Normal activation = 0.54, $p$ = 3.56E-02) and *MYOF* (PULM activation = 1.66, Normal activation = 0.12, $p$ = 3.56E-02). All four of these genes have been implicated in pulmonary carcinoma via a variety of mechanisms. *CD44* and its multiple splice variants are involved in a variety of cancer mechanisms, [63] including angiogenesis via the ERM-VEGFR axis, cell proliferation via the ERM-VEGFR-MAPK axis, metabolic shift in cancer cells via the SRC-AKT axis, and cancer cell invasion via the HGF-MET-PI3K-AKT axis. Pastushenko et al.[64] found that loss of function in *FAT1* promotes EMT (Epithelial to Mesenchymal Transition) and subsequent metastasis in lung cancer. Parimon et al.[65] found that *SDC1* (syndecan-1) controls miRNA packaging in exosomes and that this is essential for lung tumorigenesis. A truncated variant of SDC1 was shown to stimulate metastasis of fibrosarcoma in a mouse model [66]. *MYOF* (Myoferlin) can induce EMT by up-regulating mesenchymal related genes such as fibronectin and simultaneously down-regulating epithelial related genes such as E-cadherin.[67]

Our analysis of canine samples identifies tissue-specific genes involved in surfactant synthesis and cancer associated genes related to EMT, exosome packaging, and angiogenesis as unique to canine pulmonary carcinoma. This is complementary to the analysis of human samples by Peng et al. [26], which identify the upregulation of surfactant proteins as unique to human pulmonary carcinoma. A more complete picture of the pulmonary carcinoma biology is obtained when the two studies are considered together.

Many examples of multi-cancer transcriptomic analyses based on data from The Cancer Genome Atlas (TCGA) exist. One representative study is that of Li et al.[68] who utilized

transcriptomic data from TCGA to build predictor models that accurately classified 31 human tumor types. Examples of large-scale multi-cancer transcriptomic analyses in the canine realm are non-existent, mainly because of the lack of canine data. We found no examples of models that simultaneously classified canine and human data for multiple cancer types.

All these described aspects of our protocol enabled precise transcriptomic characterization in the form of a unique module expression profile for each canine cancer studied (Fig 2). This rendered it possible to: (i) develop a classification model valid for 5 cancer types across two species, (ii) focus on novel targets for drug discovery, and (iii) biologically interpret the data.

A demonstration of classification model performance is shown in the ROC plots of Fig 4. High AUC (0.99) and sensitivity (0.99) averages are obtained for specificity fixed at 0.7, when the classification models are built with one-half of the canine internal data for PULM, MEL, OSA, BLSA, and TLSA, and tested on the other half (Fig 4A and 4B). The same is true when the classification models, built with our entire internal canine data set, are tested on external canine and human data of the same cancer type (see Fig 4C–4F). Interestingly, even when the human data is seeded with cancer types different than those used to train the models, e.g., Glioblastoma, Adenocarcinoma of the colon, infiltrating duct carcinoma of the pancreas (see S4 Fig), respectable ROC statistics are obtained as seen in Fig 4G–4H (average AUC, sensitivity, and specificity = 0.91, 0.91, and 0.70 respectively). The demonstrated transferability of classification models from canines to humans enforces the idea that tumor biology and biomarker candidates discovered in canines may translate to human medicine.

The ROC statistics, however, decay as the model tests go from canine internal, to canine external, to human external, to human external seeded with cancer types other than those used to build the models. This is to be expected as the external data diverges from the data used to build the models. In addition, the external data is generated under different laboratory conditions than our internal data and these performance characteristics depend, to a large extent, on the nature of the samples used to generate data in these external studies (see S3 and S4 Figs for these data) and the nature of the internal samples used to construct our classification models (see S1 Fig for details).

For example, in the case of melanoma we tested our classification model on the external canine data of Blacklock et al. [69] (GSE129750) which was generated from samples of primary and metastatic tumor sub-types, whereas our classification model was generated from primary tumor data only. This may contribute to the lower performance seen in Fig 4C (blue ROC curve, MEL AUC = 0.92) for tests using the external canine data as compared to Fig 4A (blue roc curve, MEL AUC = 0.97) for tests using the internal canine data.

In another example, we tested our B-cell lymphoma classification model on the external human data of Péricart et al. [70] (GSE120104) and Marques et al. [71] (GSE74266). The B cell human lymphomas in these data sets contain human DLBCL lymphomas of subtype ABC (Activated B cell lymphoma) and GCB (Germinal center B cell lymphoma). Our model was trained on canine DLBCL (not otherwise specified). Even if we assume that our DLBCL data is ABC, GCB, or a mixture of both, it is known that ABC/GCB genes are not strictly conserved between dogs and humans [11]. This may explain why our B cell model performs worse as seen in Fig 4E (cyan ROC curve, BLSA AUC = 0.84) for tests using the external human data as compared to Fig 4C for tests using external canine data (cyan ROC curve, AUC = 0.98).

When we seed the human data set with data from cancer types other than those used to build the models, the AUC and sensitivity (at fixed specificity of 0.7) declines for all cancer types (except PULM, Fig 4G and 4H). The average AUC of 0.95 in Fig 4E and 4F (unseeded case) reduces to 0.91 in Fig 4G and 4H (seeded case). This trend would surely continue for additional data seeded into the test set. At some point, the gene co-expression modules and the classification models based on them need to be recalculated using all data in the expanded data set.

Ideally, transcriptomic data for all cancer types (hundreds) and non-cancer diseases (many thousands) should be collected, gene co-expression modules calculated, and classification models built therefrom. If the current computations on the 5 cancers studied in this work are any indication, the module expression profiles in the expanded case should be unique to each cancer, and the classification models built therefrom should be sensitive and specific to each cancer across the whole range of cancers considered in the expanded set. The success of our models in classifying PULM, MEL, OSA, TLSA, and BLSA, among 10–15 additional cancers considered, bodes well for these larger computation. See Li et al.[68], Newton et al.[72], and others [73,74] as examples of efforts similar in scope to those currently discussed.

Aside from their potential usefulness as cancer biomarkers, many of the module genes are understudied as drug targets [75] and provide novel opportunities for drug discovery.

We used the Pharos interface to the IDG (Illuminating the Druggable Genome) [40] to categorize our module biomarkers and found that most were in the "Tbio" class, with a few classified as "Tdark" or "Tclin/Tchem". "Tbio" and "Tdark" targets have no known small molecule activities and are therefore underexplored and novel drug targets. See Fig 12 for the identification of these underexplored drug targets (1 asterisk = Tbio, 2 asterisks = Tdark).

There is significant overlap between our module genes and the genes in COSMIC and OncoKB. Genes associated with a particular cancer type that overlap with COSMIC means that there have been found somatic mutations in that gene for samples of that cancer type. Genes that overlap with OncoKB means that they are bona fide oncogenes. See Fig 12 for the identification of these COSMIC or OncoKB genes (pink boxes = COSMIC, bold border boxes = OncoKB.

The target priority/novelty, COSMIC, and OncoKB information is overlayed onto Fig 12.

High priority "Tbio" and "Tdark" targets are appended with one or two asterisks respectively in Fig 12. COSMIC and OncoKB overlaps are delineated in Fig 12, by either a pink background for COSMIC overlap, or a bold surrounding box for an OncoKB overlap.

The information in Fig 12 can be used to pick a subset of high priority, novel drug target genes. One way to do it is to focus specifically on COSMIC and OncoKB genes, because they are known to be associated with the cancers studied. Then for these genes focus on the Tbio and Tdark targets, those for which there are no active small molecule chemistries and are understudied as drug targets. Genes satisfying these constraints are: *CD44*, *FAT1*, *SDC1*, and *MYOF* for PULM, *SPPL2A*, *CHL1*, *AP4E1*, *SECISBP2L*, and *COPS2* for MEL, *COL16A1* for OSA, and *MDN1* and *CD22* for BLSA. Some details are instructive here. For example, *FAT1* (associated with PULM in this work) is found in COSMIC and is a known oncogene on this list but strangely enough it is understudied as a drug target. Its Pharos target class is "Tbio" which means that there are no small molecule chemistries that show activity against *FAT1*. *FAT1* plays important roles in cell migration and cell-cell contact. It has been reported to be a tumor suppressor [76] and a tumor promoter [77] in different biological contexts.

*AP4E1* (associated with MEL in our work) is found in COSMIC which means that it does exhibit mutations in melanoma. *AP4E1* encodes a protein that plays important roles in the secretory and endocytic pathways which are important for cell communication in the tumor microenvironment [78]. *AP4E1* has a Pharos target class "Tdark", which means that essentially nothing is known about this target and there are no known active chemistries.

*COL16A1* (associated with OSA in our work) and *MDN1* (associated with BLSA in our work) are both found in COSMIC. *COL16A1* functions to maintain the integrity of the extracellular matrix and is a "Tbio" target, while *MDN1* functions as a transporter for ribosomal proteins and is a "Tdark" target.

One may pick a lower priority, higher risk, list of novel drug targets by only requiring that they are "Tbio" or "Tdark" targets irrespective of their membership in COSMIC or OncoKB.

This leads us, for example, to a target such as *KDELR2* for OSA. *KDELR2* is required for normal vesicular trafficking through the Golgi. Another example is *WDFY4* for BLSA which is involved in autophagy.

It is apparent from the examples above that many of these understudied proteins are involved in vesicle/protein trafficking or play supporting roles in signal transduction.

From the standpoint of biological interpretation, our approach integrates standard gene expression analysis with gene co-expression module elucidation. Within a module, genes exhibit similar activation patterns across the sample set (Fig 2) and so are likely to share common annotations, representing the underlying biology of the cancer to which the module is associated. [79] The modules are used to infer common pathway membership (via correlated expression) of a set of genes, without knowledge of the pathways. Annotation analysis, for the genes of each module, done independently of the genes in other modules, facilitates biological interpretation as evidenced by the block diagonal nature of the annotation matrix shown in Fig 5. From this block structure, annotation themes (biological interpretation) are readily seen. For example, as discussed previously, module 1 annotations for pulmonary carcinoma focus on surfactants and vesicles, module 2 annotations for melanoma focus on cytoskeletal organization, module 3 annotations for osteosarcoma focus on bone components and processes, module 4 annotations for B-cell lymphoma focus on ribosomes, and module 5 annotations associated with T-cell lymphoma focus on cell signaling, second messenger systems, and epigenetic control. The co-expression module analysis leads to an interpretation of cancer as an activation pattern of gene modules each representing different units of biology that come into play.

The demonstrated transferability of our classification models for canine cancer to the human case, the clear biological interpretation of the modules, and the fact that many biomarker genes in these modules are understudied as drug targets, are important results. It means that the gene modules, their component genes, and associated biological interpretation are likely valid for both canine and human cancer studied here. We propose that biomarker candidate genes discovered in this study (Fig 12) which work so well in classifying both canine and human transcriptomic cancer data be further validated by determining if the proteins they code for exist in tissue and blood of canines. Existence of the proteins in tissue, especially those that are understudied as drug targets, provides actionable targets for future research on canine cancer therapies and tissue-based canine diagnostics. Existence of said proteins in blood provides a basis for future research on non-invasive blood-based canine diagnostics. Therapies and biomarkers that are successful in these canine studies should then be tested in the human clinic. Since we have done the up-front work in this study to show the transferability of canine cancer genes to the human case with respect to the cancers studied, therapies and biomarkers derived from them that are successful in the canine clinic stand an excellent chance of success in human clinical trials.

## Methods

### Tissue and blood specimens

We obtained from the Pfizer-CCOGC Biospecimen Repository [29] (Fig 1A) 12 frozen tumor and 3 frozen normal canine sample sets for pulmonary adenocarcinoma, melanoma, and osteosarcoma. For B-cell lymphoma we selected 15 frozen tumor and 3 normal samples, and for T-cell lymphoma 9 frozen tumor samples and 3 matched normal canine samples. All in all, 75 samples were chosen. The normal and tumor samples were taken from the same dog. The normal samples were extracted from the vicinity of the tumor site (but far enough away so as not to include neoplastic cells). See S1 Fig and S1 File for details regarding the tumor sample

histologies and S2 Fig regarding normal sample histologies. The Pfizer-CCOGC Biospecimen Repository contains more than 60,000 samples from ~1,800 dogs using rigorously controlled standard operating protocols from seven veterinary teaching hospitals. Samples were originally harvested from treatment-naive patient canines. We used tumor samples acquired from pet dogs within the context of routine veterinary cancer care because we believe that they are a useful and underutilized adjunct to the current platform of animal-based systems used to model human cancer.

### RNA extraction and quality control

One tissue biopsy for each sample was used and from it we extracted 2 μg of RNA using an RNeasy Fibrous Tissue Mini Kit (cat No./ID:7404) [80]. We then determined RNA integrity using the Agilent 2100 Bioanalyzer [81] with RNA 6000 Nano Kit [82]. The RNA Integrity Number (RIN) of the samples ranged between 7 and 10, with RNA Integrity Number (RIN) > 7 suitable for analysis [83]. See S2 File for details.

### RNA-Seq experiments

Each RNA sample (75 altogether) from the single tumor biopsy was divided into 3 replicates and each replicate was used to make 3 cDNA libraries for sequencing. Each of the 3 libraries was loaded onto 3 different lanes on the flow cell that goes into the sequencer. 225 libraries altogether were sequenced. The replicates allow for estimation of within sample variability of gene expression which allows us to then make inferences between samples of the same cancer type or between samples of different cancer types [84]. We used the NebNext Ultradirectional RNA library kit [33] for preparation of RNA libraries for Illumina sequencing. This method used two, iterative Oligo-dT binding steps to enrich mRNA from total RNA. 225 Illumina library preparations were made from mRNA. RNA-Seq experiments were performed at ~ 150X coverage using the Illumina HiSeq 2500 instrument (Fig 1B) [34]. (2*125bp paired end reads, an average of 45 million reads per sample).

### RNA-Seq data processing

The sequence reads (Fig 1B) were produced as 1,396 fastq files containing 125 base-pair paired end reads. All primary processing of the sequencing data was performed on the National Institutes of Health Biowulf High Performance Computational system (https://hpc.nih.gov/systems/). FastQC (https://www.bioinformatics.babraham.ac.uk/projects/fastqc/) was used to assess sequence quality prior to further processing. We then aligned the fastQ reads to the canFam3.1 [83] reference genome using the *STAR* [85] aligner. The HTSeq-count script from HTSeq [86] was used to analyze the read alignments and assign reads and read counts to genes by referencing the CanFam3.1 GTF annotations file from Ensembl.

The result of these manipulations is a gene by sample count matrix, which we denote by the symbol *C*, with matrix elements given by $C_{g,s}$. The row dimension of this matrix spans 24,580 canine genes, *g*, and the column dimension spans 225 samples, *s* (Fig 1C). The read count matrix is given in S3 File.

Each column of the raw count matrix, **C**, was scaled to account for slightly different number of reads for each sample. Rows (delineated by genes) and columns (delineated by samples) were further processed as follows:

- We retained only rows for which the canine official gene symbol has an associated human homolog. [87]

- We retained only rows (genes) for which the number of counts accumulated for the gene is greater than 25 for at least 185 of the 225 samples.

- We retained only rows (genes) for which the count variance of the gene across the 225 samples is in the upper half of the sorted (decreasing in value) list of variances for all of the genes. [88]

- Columns (samples) were averaged across technical replicates and only the sample replicate averages were retained.

The final matrix contains 4,115 genes (rows) and 75 sample replicate averages. The matrix was then centered and scaled using a Z transformation (function **scale** from the R base package [89]):

$$Z_{g,s} = \frac{(C_{g,s} - \langle C \rangle_s)}{\sigma_s} \tag{1}$$

where $<C>_s$ is the average read count across all genes for samples $s$ in $C$, and $\sigma_s$ denotes the standard deviation of the average read count. The Z matrix is provided in S4 File.

## Gene co-expression modules

Initial gene sets were generated by applying standard clustering to the Z-transformed matrix (function hclust from the R package fastCluster [90]). The clusters were generated such that the number of genes per cluster was $> 10$ and $< 100$, a range consistent with the approximate number of genes in a KEGG pathway [31]. 49 clusters (gene sets) were obtained. The number of clusters was expanded to 23,569 by adding 100 randomly selected genes to each cluster, 481 times.

These 23,569 gene clusters were used as input into the R package *eisa* which utilizes the Iterative Signature Algorithm (ISA) [30] for gene set refinement and generation of a finalized set of gene co-expression modules associated with the entire Z-score matrix. (Fig 1D and S5 File). These modules were calculated with our lab local canine data and from henceforward, the identities of the module genes are fixed.

## Module activation profiles

The activation of gene, $g$, associated with cancer sample, $s_c$, is:

$$A_{g,s_c} = Z_{g,s_c} \tag{2}$$

where the Z score matrix elements, $Z_{g,s_c}$, are given by Eq (1). The activation of gene, $g$, associated with cancer, $c$, is the average activation of gene $g$ across the set of samples representing cancer $c$,

$$A_{g,c} = \frac{1}{N_c} \sum_{s_c=1}^{N_c} A_{g,s_c} \tag{3}$$

where $N_c$ is the number of samples representing cancer $c$. The activation of module, $m$, associated with sample, $s_c$, is the average activation across all genes in module $m$ for sample $s_c$ and is given by:

$$A_{m,s_c} = \frac{1}{N_m} \sum_{g=1}^{N_m} A_{g,s_c} \tag{4}$$

where $N_m$ is the number of genes associated with module $m$ and the $A_{g,s_c}$ elements are defined

by Eq (2). The activation of module $m$ associated with cancer $c$ is the average of the $A_{m,s_c}$ across all samples, $s_c$, representing cancer, $c$, and is given by:

$$A_{m,c} = \frac{1}{N_c} \sum_{s_c=1}^{N_c} A_{m,s_c} \tag{5}$$

where the $A_{m,s_c}$ elements are defined by Eq (4). Module activations for the various cancer types are shown in Figs 2 and 3A.

The reference state for each cancer type is a matched normal sample taken from the same dog in the vicinity of the tumor site. The activation of module $m$ for the normal case is given by:

$$A_{m,n} = \frac{1}{N_n} \sum_{s_n=1}^{N_n} A_{m,s_n} \tag{6}$$

where $n$ is the normal sample index, $N_n$ is the total number of normal samples (3 altogether for each cancer), $s_n$ designates normal sample, the sum runs over all normal samples associated with the cancer, and

$$A_{m,s_n} = \frac{1}{N_m} \sum_{g=1}^{N_m} A_{g,s_n} \tag{7}$$

where $N_m$ is the number of genes associated with module $m$, $g$ is the gene index, and $A_{g,s_n}$ is the activation of gene $g$ for normal sample $s_n$. Module activation for the normal samples is given in Fig 3B.

The statistical significance of all module (Eq (5)) and gene (Eq (3)) activations relative to normal activation (Eq (6)) were calculated using either a standard two-tailed T test [91] or the non-parametric Wilcoxon rank sum test [92]. The T test was used if the distribution of module activations across tumor samples and normal samples was determined to be normal according to the Shapiro-Wilks test. [93] If either distribution was determined not to be normal, then the Wilcoxon rank sum test was used. The resulting $p$ values were then multiple testing corrected using the method of Benjamini-Hochberg. [94]. The computed $p$ values are given in Fig 3C.

The module activation profile for a given cancer, $P_c$, (Fig 3A rows) is the set of module activations associated with each cancer:

$$P_c = \{A_{m,c}\}; m = 1 - N \tag{8}$$

where N is the total number of modules determined from the ISA analysis (in our work N = 5). For the normal activation profiles, Eq (8) applies but with the index $c$ replaced by $n$ (Fig 3B rows). After the profiles were computed, the primary module of each cancer was determined as the highest significantly expressed module, statistically different between cancer and normal states. The primary modules were determined to be modules 1, 2, 3, 4, and 5 for pulmonary carcinoma, melanoma, osteosarcoma, B-cell lymphoma, and T-cell lymphoma, respectively (Fig 3A row and column labels for colored entries, e.g., PULM, primary module = 1, color blue, etc).

We used the R function boxplot() (R package gplots) to create a plot of the distributions of primary module gene activations for each cancer and matched normal (Eq (3)) sample shown in Fig 3D. We used the R function pca() (R package PCAtools) to run a PCA analysis of the module by sample activation matrix (Eq (4)). The result of this analysis was used as input to the R function biplot() (also PCAtools) to construct the PCA plot shown in Fig 3E.

## Predictive cancer models: Nearest neighbor models

Using Eq (8) the cancers can be represented by point locations in a 5-dimensional cartesian space with axes defined by module activations computed with Eq (5). Each cancer occupies a different position in this space (idealized in Fig 1F as large boxes). New transcriptomic data from an external source can be "classified" by using its gene expression values to compute the module activations (remember the module genes were determined using lab-local internal canine data and are fixed), then positioning the new sample in module activation space (small boxes, Fig 1F). The distance between the new sample point and each of the cancers is computed, using:

$$D_{c, s_{external}} = \sqrt{\sum_{m=1}^{N} \left(A_{m,c} - A_{m, s_{external}}\right)^2} \tag{9}$$

where c is the cancer-type index, $s_{external}$ represents the external sample to be classified, $m$ represents module, the summation runs from 1 to the total number of modules $N$, $A_{m,c}$ is the module activation of the cancers as calculated by Eq (5) and $A_{m, s_{external}}$ is the module activation of the new sample as calculated by Eq (4). The sample is then identified as the cancer to which it is closest to. Before any such classification can be performed, the external data and the internal data must be simultaneously normalized so that the data sets, typically from different laboratories, have the same dynamic range. To do this we apply quantile normalization using the function normalize.quantiles() from the R package preprocessCore version 1.46.0.

## Model validation using internal canine data

This distance model was validated using lab-local canine data by random subsampling cross validation. In this method the sample set is split in half (by choosing at random) to create a training and test set each consisting of 30 tumor samples across the 5 cancer types PULM, MEL, OSA, BLSA, and TLSA for each set. Modules are computed using the training set samples. The module activations for each training set cancer are computed using Eq (5) and the training set cancers are positioned in module space (Fig 1F upper, large squares). Using Eq (4) the module activations are then calculated for each test data sample (using the modules determined from the training set, but the test set gene expression data) and each test sample is positioned in module space (Fig 1F upper, small squares, each colored according to cancer type). For each training set cancer position, a list of test sample positions is generated, and sorted increasing by distance (using Eq (9)) from the training cancer positions (Fig 1F upper, numbered small squares, distance from melanoma, MEL, as example). This produces 5 sort orders, one for each training set cancer. For each training cancer, we identify the locations of the test cancer samples of the same cancer type in the sorted set of test conditions (Fig 1F lower) and from that construct a ROC plot tracking the true positive rate versus the false positive rate (Fig 1G). Evaluation metrics are then computed, these are area under the curve (AUC), sensitivity, and specificity. The procedure is repeated 200 times and the finalized evaluation metrics are computed as averages over the individual runs.

## Model validation using external data

When validating the model using external data, we use all internally generated canine data (4,115 genes and 60 tumor samples across 5 cancer types) to compute the gene co-expression modules. Then we proceed as described above, but with two differences:

1. The module genes are fixed and not recalculated during random subsample cross-validation.

2. We impose the constraint that each cancer is represented by the same number of samples in the training and test sets.

3. Human external data for cancers other than the ones studied are added to the data set as background negatives

We selected external data from NCBI-GEO (https://www.ncbi.nlm.nih.gov/geo/) [95] and the Genomic Data Commons Data Portal (GDC, https://portal.gdc.cancer.gov/). The various NCBI-GEO and GDC data series selected are shown in S3 and S4 Figs, respectively. Data from these sets was selected, as much as possible, according to the following criteria to make it as consistent as possible with our own internal data:

1. The data is associated with primary tumors.

2. The data is associated with tumors from treatment naive individuals.

3. Where possible the data must be RNA-Seq.

4. Where possible we used data from which gene counts were computed using STAR [85] alignments.

We perform random sub-sample cross validation to test our models. When testing a model, we pool all data together, then split it evenly into training and test sets, requiring that the number of positive cases (for any cancer we are focusing on) is the same in training and test sets. This avoids ambiguity when interpreting the ROC statistics. Using the fixed module genes (determined from our internal canine data), we use the training data in each trial to recalculate the positions of each cancer in module activation space. The rest proceeds as described above for validation using internal data. We performed 4000 trials of random subsample cross validation when using the external data. The reason for this large number of trials is that each trial uses only a subset of the data available, to enforce the constraint that the number of positive cases for each cancer type be the same. Therefore, we must increase the number of trials to effectively use all the data available. The converged ROC results are shown in Fig 4.

## Module annotations

Genes sets from individual co-expression modules were submitted, one set at a time, to the David bioinformatics resource [35] functional annotation tool, to find associated KEGG [31] pathways and GO [32] terms. We used Benjamini-Hochberg [36] corrected $p$ values to guide selection ($p < 0.05$) of annotation terms. We organized the results of this search in the form of an annotation matrix with rows enumerated by the different annotations and columns enumerated by modules and their associated cancer types. Matrix elements were populated with 1 or 0 depending on whether the annotation is or is not associated with the module. See S6 File for the full annotation matrix. Fig 5 shows a smaller version of this matrix, including only annotations with $p < 0.005$.

## Juxtaposition of annotations, genes, modules, cancer types, and biomarker selection

For each module and its' associated genes and cancer type, we performed the following operations:

1. We sorted genes decreasing by their average activation across samples representing the cancer.

2. We extracted annotations associated with the module genes from S6 File.

3. We flagged individual genes associated with each annotation.

4. For each gene we gathered evidence from the Comparative Toxicogenomics database (CTD) indicating an associating between the module gene and the cancer

The result of these manipulations is contained in S7 File. The organization of the tables in this file is as follows:

1. Module genes are in column A and their associated cancer and normal activations (Eqs (3) and (6)) in columns E and F, respectively. Module genes are sorted decreasing by their cancer activation values.

2. Annotations associated with the module start in column H and are arrayed horizontally in row 3.

3. BH corrected $p$ values associated with the annotations are shown in row 1 starting in column H.

4. Matrix elements $(i, j)$ start in row 7, column H and contain a 1 if the gene on row i is associated with the annotation in column j, and a 0 if it is not.

5. Row 4, below the annotations tallies the total number of genes in the module associated with the annotation. The elements of this row are sorted decreasing (horizontally) to position the most important annotation towards the left.

6. Row 5 tallies the total number of genes with activation $> 1.5$ associated with the annotation.

7. The gathered evidence associating each gene in the module to the cancer is in columns B (Pharos IDG class [40]), C (CTD [41] inference score), and D (CTD number of references). The queries used to pull the CTD evidence are shown in row 3, columns A-F. See the next section for descriptions of the Pharos and CTD evidence.

8. Column G contains BH corrected $p$ values for gene activation across cancer conditions (column F) compared to gene activation across normal conditions (Column E). $p$ values are colored pink if they are $< 0.05$.

Smaller versions of this matrix (Figs 6–11) were constructed for each cancer type, but only including those annotations and genes specific to the cancer type and only those genes with activation $> 1.5$. The activation cut-off value of 1.5 is used, because gene activations are based on Z scores and activation values $> 1.5$ (relative to background) are significant at the 95% confidence level. All the genes from Figs 6–11 were gathered into one place and are presented in Fig 12 as a finalized set of biomarker candidates.

## Biomarker link to literature and target novelty

The module genes were submitted as queries to the comparative toxicogenomics database (CTD) [37] to find evidence supporting their link to the cancers they are associated with. The evidence has two components:

1. The inference score: [41] A function of the relationship that the gene has to a chemistries known to influence other genes that are part of the cancer mechanism. Inferences scores are $\geq 1$, with larger values indicating a stronger gene-cancer connection.

2. The number of published articles documenting association of gene to cancer.

See Figs 6–11 and S7 File, columns C and D.

The module genes were also submitted to Pharos, a web-based interface to the Target Central Resource Database (TCRD). [40] This database was assembled by the Illuminating the Druggable Genome (IDG) program and categorizes proteins according to the level of knowledge associated with each. There are four broad Target Development Levels (TDL) or IDG classes: Tclin, Tchem, Tbio, and Tdark. Tclin targets are those associated with approved drugs of known mechanism of action, Tchem targets are associated with small molecule activities ($< 30$nM) in ChEMBL, [96] Tbio targets do not have known drug or small molecule activities, but they do have associated gene ontology terms or a confirmed OMIM [42] phenotype, and Tdark targets are those that are essentially unstudied. These target development levels are shown in column B, Figs 6–11 and in the tables included in S7 File. In Fig 12, Tbio and Tdark targets are appended with 1 or 2 asterisks respectively.

To prioritize target novelty of biomarkers in terms of druggability, a combination of CTD, Pharos, and expression information is used. In the case of targets associated with PULM, where valid normal samples exist (normal samples from the same tissue type as the tumor samples), the following criteria are used:

1. The target activation (Z score) $> 1.5$.

2. The target $p$ value for tumor expression vs. control is $< 0.05$.

3. The Pharos target development level is either Tbio or Tdark.

4. For Tbio targets, the CTD inference score is $< 10$ and the CTD number of references $< 5$.

In the case of MEL, OSA, BLSA, and TLSA, $p$ values are ignored because valid normal samples for these cancers are not available (criteria 2).

The activation cut-off value of 1.5 is used above, because gene activations are based on Z scores and activation values $> 1.5$ (relative to background) are significant at the 95% confidence level.

These criteria reduce the number of targets to a manageable level for future exploration/ studies. If resources are available, one may relax the criteria to consider additional targets and, if not, one may use more stringent criteria to consider fewer targets.

## Supporting information

**S1 Fig. Canine tumor samples and assigned histology used in this study.**
(TIF)

**S2 Fig. Normal sample gene markers and assignment to tissue type.** Normal samples were from the same dog as the tumor samples and were taken as near the tumor site as possible but far enough away so as not to include any neoplastic tissue. The top 200 expressed genes from each normal sample were submitted to the Tissue Specific Expression Analysis Tool (TSEA) [44] to identify the origins of the normal samples.
(TIF)

**S3 Fig. Geo data sets used to test canine and human classification models developed on internal canine data.**
(TIF)

**S4 Fig. Genomic data commons data sets used to test human classification models developed on internal canine data.**
(TIF)

**S1 File. Biospecimens.**
(XLSX)

**S2 File. RNA quality.**
(XLSX)

**S3 File. Raw count matrix.**
(TXT)

**S4 File. Z matrix.**
(XLSX)

**S5 File. Gene modules.**
(XLSX)

**S6 File. Annotations: Full.**
(XLSX)

**S7 File. Annotations: Gene detail.**
(XLSX)

**S8 File. Tumor, stroma, and breed purity study.**
(DOCX)

**S9 File. Tumor, stroma, and breed purity study data.**
(XLSX)

## Acknowledgments

We would like to thank Christopher P. Austin (Director, NIH/NCATS), Anton Simeonov (Director, NIH/NCATS/DPI), and Don Lo (Director NIH/NCATS/DPI/TDB), for their support to make this research possible. **Biospecimens**: We would like to thank the CCOGC for their kindly donation of canine tumor samples used in this work. **Pathology**: We would like to thank Dr. Lauren J. Harris for performing the histological analysis of the samples used in this study. **Disclaimer**: The content of this publication does not necessarily reflect the views or policies of the Department of Health and Human Services, nor does mention of trade names, commercial products, or organizations imply endorsement by the U.S. Government.

## Author Contributions

**Conceptualization:** Gregory J. Tawa, David Gerhold, Matthew Breen, Gurusingham Sittampalam, Amy K. LeBlanc.

**Data curation:** Gregory J. Tawa, John Braisted, Christina Mazcko, Matthew Breen.

**Formal analysis:** Gregory J. Tawa, John Braisted.

**Funding acquisition:** Gregory J. Tawa, Amy K. LeBlanc.

**Investigation:** Gregory J. Tawa, Matthew Breen, Amy K. LeBlanc.

**Methodology:** Gregory J. Tawa, Matthew Breen.

**Project administration:** Gregory J. Tawa, Christina Mazcko.

**Resources:** Gregory J. Tawa, Gurmit Grewal, Christina Mazcko, Amy K. LeBlanc.

**Software:** Gregory J. Tawa.

**Supervision:** Gregory J. Tawa.

**Validation:** Gregory J. Tawa.

**Visualization:** Gregory J. Tawa, John Braisted, Amy K. LeBlanc.

**Writing – original draft:** Gregory J. Tawa, Amy K. LeBlanc.

**Writing – review & editing:** Gregory J. Tawa, David Gerhold, Gurmit Grewal, Christina
Mazcko, Matthew Breen, Gurusingham Sittampalam, Amy K. LeBlanc.

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
