## [Decision Letter · Decision Letter 0]

24 Mar 2021

Dear Dr. Tawa,

Thank you very much for submitting your manuscript "Transcriptomic profiling in canines and humans reveals cancer specific gene modules and biological mechanisms common to both species" for consideration at PLOS Computational Biology.

As with all papers reviewed by the journal, your manuscript was reviewed by members of the editorial board and by several independent reviewers. In light of the reviews (below this email), we would like to invite the resubmission of a significantly-revised version that takes into account the reviewers' comments.

We cannot make any decision about publication until we have seen the revised manuscript and your response to the reviewers' comments. Your revised manuscript is also likely to be sent to reviewers for further evaluation.

Sincerely,

James Gallo

Associate Editor

PLOS Computational Biology

Florian Markowetz

Deputy Editor

PLOS Computational Biology

Reviewer's Responses to Questions

**Comments to the Authors:**

Reviewer #1: This is a straight forward paper and I have no issue with the methodology or the conclusions. However, it is purely observational with no functional data or analysis beyond description. If the editor and journal are fine with this, then publication is appropriate.

Reviewer #2: The authors describe a gene expression study based on RNA sequencing of 225 tumor and normal samples from 12 tumor samples and 3 matched controls for each of five canine cancer types: pulmonary carcinoma, melanoma, osteosarcoma, and B- and T-cell lymphoma from 60 dogs. They use a unique unsupervised clustering method to identify five co-expression modules and cancer type-associated expression profiles that also support shared cancer

biology between canines and humans for these cancer types. This is an important and well-designed study in the field of canine and comparative transcriptomics not only for what it reveals of transcriptomic similarities between five canine and human cancer types, but also due to the framework through which it rigorously establishes those similarities. Mostly minor needs are highlighted for revision to improve clarity and understanding of the value of the approach the authors have taken in the context of standard approaches and existing knowledge.

1. Minor comments:

a. Although this study is focused squarely on transcriptomics and expression, it would nonetheless be helpful to add some perspective in the Intro and Discussion on the genomic (mutation-based) underpinnings at play in canine and comparative cancer genomics studies and that hold relevance for downstream transcriptional differences and/or possible tumor subtypes. This is particularly true since the genomic underpinnings of some of these tumor types are increasingly well-understood. Are there any possible connections, for example, between SETD2 mutations in osteosarcoma and any of the genes or expression modules discovered here? Any reason that possible similarities or differences exist between canine and human linked back to genetic underpinnings?

b. Is there no value at all in performing more traditional gene expression analysis such as assessment of individually significantly differentially expressed genes in each tumor type, exploration of clusters based clinical features, etc.? It seems like it would be particularly effective to do that here since you have such a great set of replicates and sampling of tumor types. This could even be described preceding the unique approach here taken to highlighting co-expression modules and their constituents. Then, the relative value of each approach could be commented on.

c. Pg. 3 - How were the samples divided into three replicates? Was this at the tissue level, analyte level, sequencing library level, etc.? Methods makes it sound like RNA was extracted from a subdivided piece of tumor, but it’s not totally clear.

d. Similar to “d,” was tumor content somehow verified in each section? How do you know that you were sequencing primarily tumor and not normal stroma and how might you account for variability in tumor content between the replicates if they are from subdivided tumor sections?

e. Pg. 3 - The first section of the Results doesn’t really seem to be an independent result of particularly advanced analysis worthy of its own segment, but perhaps should be a comment in the following section when addressing various elements of the cohort composition.

f. Pg. 3 – Recommend retitling the second Results section as “Five gene co-expression modules represent coordinated activation patterns of gene aggregates associated with the 5 types of cancer” or something similarly more declarative rather than the current title that’s somewhat equivocal.

g. Pg. 3 - It’s not immediately clear how we went from a 24,580 x 225 sample matrix to a 4,115 x 75 sample matrix.

h. I noticed a handful of run-on sentences such as the following on page 3: “We determined the expression profile of each of these modules across the samples, these are shown in Fig. 2.”

i. There’s a lot of unusual nomenclature in here that can be difficult to relate back to the underlying biology and/or is perhaps even not specific enough, so it quickly gets confusing (at least for me). Can any of this be simplified or reworded or more precisely defined? For example, “coordinated activation patterns,” “gene aggregates,” “coupled transcriptional events associated with multiple overlapping and intertwined cancer mechanisms,” “uniqueness property,” and even “co-expression module” are not immediately well-defined or intuitively understandable.

Reviewer #3: This manuscript provides novel insights into possible shared transcriptional pathways between human and canine malignancies as well as highlighting the potential utility of using spontaneous canine malignancies as a model for human cancers. The authors use RNAseq data from 5 different cancer types to find genes which form co-expression modules and later apply these modules to human cancers to show cross-species utility of such findings.

Main Text:

It is unclear where the “matched controls” come from, what type of tissue is used, and if it is from the same animal (may be in supplementary tables we don’t have access to?). This is of specific importance as it determines whether the proposed biomarkers are relevant in determining cancerous vs non-cancerous tissue or, alternatively, simply identifying what type of tissue has been collected. Comparing each module activation against the average activation across all normal tissue types of all origins is possibly just selecting for tissue-specific transcription rather than cancer-related gene expression. Analysis of module activity should include classification of normal tissue samples as well as it relates to tissue type as this is critical in presenting this data as cancer-specific biomarkers.

Figures:

Figure 2: This figure is difficult to interpret, and this data may benefit from being displayed as a heatmap with samples as columns and genes as rows to better appreciate the grouped activation. If this stays as a line graph, a key should be included for the genes each color represents.

Figure 3: This data may be better presented as a boxplot or with individual data points showing so that the distribution of such values is better represented rather than the mean and standard deviation. This data should also be derived from appropriately labeled samples as mentioned below.

Figure 5: This table should include the number of genes within the model as well as the number of genes within the defined set. Tables 2-6 show that many of the “enriched” gene sets in each module only contain a single gene from the module, which limits the utility of using such gene set annotations to aid in defining the module. It may be more useful to evaluate the same data sets taking into account the overall activation of each genes using GSEA (https://www.gsea-msigdb.org/gsea/index.jsp) as it seems a single, barely activated gene should constitute gene set enrichment (for example, Table 2 – PTGIS in oxidation-reduction process). Such analysis should also account for the fact that the number of each genes in the set is limited by looking only at the intersection of canine and human genes, greatly reducing the number of genes within each set.

Supplementary Data: Major reservations for this manuscript come largely from the information revealed in the supplementary data detailing the misclassification of T-cell lymphoma samples and the decision to present the entirety of the main manuscript with these samples included in the incorrect group. The authors suggest that “the module activation profiles are essentially unchanged” though activation of module 4 in the TLSA samples nearly halves upon proper classification while module 5 activation within the same group increases by roughly 25% in the same group. Reclassifying the samples greatly increases the specificity of module 5 for TSLA. These samples also make up about 25% of the total samples used in this analysis, so it is disingenuous to conclude that the rest of the manuscript should not be altered to reflect the appropriate classification of such samples.

**Have all data underlying the figures and results presented in the manuscript been provided?**

Reviewer #1: Yes

Reviewer #2: Yes

Reviewer #3: **No: **CanineCommons website was down at the time of access.

PLOS authors have the option to publish the peer review history of their article (what does this mean?). If published, this will include your full peer review and any attached files.

Reviewer #1: No

Reviewer #2: No

Reviewer #3: No
---

## [Decision Letter · Decision Letter 1]

11 Jun 2021

Dear Dr. Tawa,

Thank you very much for submitting your manuscript "Transcriptomic profiling in canines and humans reveals cancer specific gene modules and biological mechanisms common to both species" for consideration at PLOS Computational Biology. As with all papers reviewed by the journal, your manuscript was reviewed by members of the editorial board and by several independent reviewers. The reviewers appreciated the attention to an important topic. Based on the reviews, we are likely to accept this manuscript for publication, providing that you modify the manuscript according to the review recommendations.

Sincerely,

James Gallo

Associate Editor

PLOS Computational Biology

Florian Markowetz

Deputy Editor

PLOS Computational Biology

[LINK]

Reviewer's Responses to Questions

**Comments to the Authors:**

Reviewer #2: The authors have done an extraordinary job of addressing all of the reviewer comments.

Reviewer #4: Tawa et al. have analyzed canine transcriptomics data in the context of existing human data to evaluate the comparative relevance of canine to human cancer. They show that tumour biology and biomarker candidates discovered in canines may translate to humans. Overall, the methods and results of this paper are well done and straightforward to follow.

Furthermore, the Authors have taken all of the reviewer's comments seriously and addressed them appropriately. I have no further comments or possible correction on the current manuscripts.

Reviewer #5: The authors developed an protocol that analyzes canine transcriptomic data in the context of existing human data to evaluate comparative relevance of canine to human cancer. They characterized five canine cancers, each exhibiting a unique module expression profile. The canine-derived models they generated successfully classified human tumors representing the same cancers. They also identified 104 biomarkers as diagnostic biomarker candidates. I only have some minor suggestions.

Figure 2. what is the unit for the y-axis?

Figure 3. It seems that most of the values in Figure 3C are significant, whereas only a few are labeled with * in D. What are the criteria for being regarded as significant?

Tables. Please explain what 0 and 1 refer to and what the red color denotes.

Reviewer #6: In this paper, Tawa, et. al., describes a study using gene expression from a series of canine tumor and normal tissue samples to identify gene expression modules that can classify tumor types in the context of both canine and human disease. I agree with the original reviewer 1 that this is a strictly descriptive paper (no functional analysis performed) and would benefit from modifying the focus of the discussion less to how these results recapitulate aspects of results already seen in other systems and more to what new science this analysis enables. Nonetheless, the approach is straightforward and if the editor and journal are fine with descriptive work, then this is fine.

The authors largely responded to the comments of reviewers 2 and 3. Find some additional minor concerns that need to be addressed below:

As a general note, all of the figures that were part of the reviewer packager were very low resolution and when they involved small text in a table is was often illegible. Please double check that you have the appropriate resolution settings when you generate your figures for publication.

The incorporation of the discussion around the common mutational landscape in canines and humans is somewhat awkward. I would recommend moving it earlier in the introduction to provide further justification for why canines are a useful model system to understand human cancer and then lead into gene expression and stay there.

Intro paragraph 3, sentence 1: clarify “these studies.” Are “these studies” the studies that compare human and canine cancer biology to validate canines as a relevant model system? Or are “these studies” studies of cancer biology and experimental therapeutics?

Be consistent in bolding Figure references in the main text. For example the first reference to Fig 1A is bolded, but then Fig 1B and on are not.

Based on your previous responses to the original Reviewer 2, Figure 1A is still misleading as to the nature of the workflow. It suggests that you have 45 separate samples for each tissue or origin that were processed to RNA, when, in fact, you had 15 separate samples with RNA extracted and then you split the RNA into three separate libraries. Please change this workflow to reflect what was actually done. Change the yellow “45” to “15.” Change “225 samples” to “75 samples” and then alter the figure to indicate that this was then used to generate technical replicates.

For Figure 3 A-C, please change the text over the yellow boxes (Module 2 for melanoma) to black to make it legible.

For Figure 5, is there any way to color code the p-value column? Toward the paper’s argument that unique signatures are activated in each cancer, multiple of the signatures associated with module 3 and module 4 have much lower p-values than thos associated with multiple cancer types.

On page 9 it appears there is an erroneous line that says “Fig 10. Continued”

On page 10 in the second paragraph of the discussion, the second line currently reads “cancer types and applicable…” and should be replaced with “cancer types and are applicable…”

Reviewer #7: The authors analyzed RNA-Seq data obtained from 225 tumor and normal canine samples and compared it to published human transcriptomic data to identify cancer-type-specific gene co-expression modules that are conserved between the two species. They then functionally annotated the modules to gain insights into biological mechanisms.

Major Comments

Novelty/originality of work

1. Dataset

All but the internal canine data has been already published.

2. Methods

Authors use a straight-forward approach that averages gene expression counts per gene list per condition using standard R packages. It is not clear to me how this is superior to other widely-used approaches for gene expression analysis such as WGCNA.

3. Biology

The authors’ observations are consistent with the work done by multiple other teams. However, what exactly are the novel targets that this analysis has discovered?

Methodology

1. Model validation to human data.

Given that the human dataset used here for validation of the modules consisted of the same 5 cancer types, the specificity of the modules to these cancer types needs to be further explored. It is crucial that the validation dataset is expanded by other cancer types to prove that these cancer-type-specific modules (M1, M3 etc) won’t match to something unexpected.

Minor points here: were all human data treatment naïve? Why did the classification model perform better on human data than on external canine data (Figure 4)?

2. Primary modules selection.

“After the profiles were computed, the primary module of each cancer was determined as the highest up- regulated module, statistically different between cancer and normal states.”

This makes me think that over-expression is the only mechanism of cancerogenesis that was considered in this study. Authors should explain why they didn’t look at the opposite trend, most down-regulated modules. For example, if a module consists of tumor suppressors, down-regulation might be a possible mechanism of tumorigenesis.

3. PCA analysis.

The PCA plot (Figure 3) shows only PC1-PC3. Perhaps I missed it, but it would be interesting to see what feature PC2 corresponds to.

Also, from my experience, sample purity is often a confounding factor in such studies. So, I would like to see scatter plots for purity (or tumor to stroma cell ratios) ~ PC1/2/3, to make sure it is not the case here.

4. Normal tissue.

The “… B and T cell lymphoma normal tissues as skin with an admixture of adipose tissue, the osteosarcoma and melanoma normal samples were mainly skin with an admixture of muscle tissue, the pulmonary carcinoma normal samples were mainly lung with an admixture of muscle tissue.”

Gene expression is known to be highly tissue specific. Given that for all but one cancer types the normal sample was not represented by the same tissue type as the tumor sample, is it sensible to select cancer-specific modules by comparing activation profiles between tumor and not-matching control samples?

5. Dog breeds.

The dataset used in this study contains RNA-Seq data on multiple dog breeds. In contrast to human genetics, purebred dogs are known to have high levels of homozygosity, which is another reason why transcriptomic studies of canine cancers are so intriguing. The authors have breed information for almost all samples in S1. In my opinion, these data should be used to stratify transcriptomic data prior to gene co-expression analysis. At least the authors should check how much variance it accounts for. Is it PC2?

6. Modules annotations.

Authors used DAVID tool for functional annotations. To increase clinical relevance of this study, it would be interesting to overlap the modules with known cancer genes from COSMIC or OncoKB.

Minor Comments

Text

1. Page 3.

“Genes in a co- expression module are of interest because such genes are thought to members of the same biological pathways or part of the same transcriptional regulatory program.”  “Are thought to be members”

2. Page 4

“Though this work is strictly a study on gene expression, the annotations provide a means for linking the most highly expressed genes for each cancer type studied with mutational events known to drive the cancers. “  Did the authors actually check that? If not, maybe it is better to move this to Discussion.

“Each RNA sample from the single tumor biopsy was divided to make 3 libraries for RNA-Seq. “  Please explain the rationale.

3. Page 6.

“Each cancer exhibits a unique module expression profile.”  This phrase sounds out of context and really stands out, please consider rephrasing.

4. Methods

“See supplementary Fig S1 and supplementary file S1 for details regarding the tumor sample histologies and location of normal samples.” – I did not find any info on location of normal samples in S1.

Figures

1. Figure 1A.

Explain clearer where the 225 samples come from and/or explain in the text how many tumor and normal samples were collected per animal.

“We prepared 225 samples for RNA-Seq analysis: 12 tumor samples and 3 normal samples in triplicate for each cancer type (Fig 1B)”.  please, specify these are technical replicates.

2. Figure 1B.

Seems unnecessary to show how RNA-Seq works in this dry lab paper. Also, since there’s no explanation in the text, better to leave it out.

3. Figure2.

As it is now, there are multiple genes (i.e. PKNOX2 from M2) that don’t change the expression between tumor and normal and hence are not informative. Consider showing a heatmap with only “biomarkers” (“genes with activation > 1.5”) and move the current version to Supplementary Figures. Also, if possible, use a more standard heatmap color scheme (red/blue).

4. Figure 3A-C.

It will help the reader a lot if you order cancer types in this table according to modules number (i.e. PULM  MEL  OSA etc)

5. Figure 3D.

Why so many outliers for M4 in BLSA normal samples?

6. Figures 5-10.

These tables should be replaced by a plot that is easier to understand, for example, boxplots showing CTD values per module.

**Have the authors made all data and (if applicable) computational code underlying the findings in their manuscript fully available?**

Reviewer #2: Yes

Reviewer #4: Yes

Reviewer #5: Yes

Reviewer #6: Yes

Reviewer #7: Yes

PLOS authors have the option to publish the peer review history of their article (what does this mean?). If published, this will include your full peer review and any attached files.

Reviewer #2: No

Reviewer #4: No

Reviewer #5: No

Reviewer #6: No

Reviewer #7: No

Figure Files:

Data Requirements:

Reproducibility:

References:

---

## [Decision Letter · Decision Letter 2]

8 Sep 2021

Dear Dr. Tawa,

Thank you very much for submitting your manuscript "Transcriptomic profiling in canines and humans reveals cancer specific gene modules and biological mechanisms common to both species" for consideration at PLOS Computational Biology. As with all papers reviewed by the journal, your manuscript was reviewed by members of the editorial board and by several independent reviewers.

The paper has been greatly improved and can be accepted once some minor comments from one of the reviewers can be addressed. 

Sincerely,

James Gallo

Associate Editor

PLOS Computational Biology

Florian Markowetz

Deputy Editor

PLOS Computational Biology

[LINK]

Reviewer's Responses to Questions

**Comments to the Authors:**

Reviewer #5: The authors have addressed all my concerns. The manuscript is ready for publication.

Reviewer #6: The authors have done a very thorough job of responding to the reviewer comments through multiple rounds of review.

Reviewer #7: The authors did a lot of work on addressing my and other reviewers’ comments. In my opinion, this manuscript has improved dramatically since the last submission. The authors improved their methodology using more appropriate subsampling strategy and added an important analysis piece on validating their results on a larger human dataset on naïve tumors of various cancer types other than the ones studied in this work. Another important type of the analysis that was performed was a comprehensive dog breed and tumor purity analysis. And, lastly, the authors annotated their gene modules with COSMIC and OncoKB. In my opinion, the large overlap with COSMIC/OncoKB genes has raised the relevance of this work.

I would also like to acknowledge how the discussion section has improved and expanded. In particular, by adding the discussion on the novelty of the targets.

Moreover, the authors have published their dataset.

Taken together, I recommend this manuscript for publishing after fixing some of the minor points that I am listing below.

Text

1. Page 3 starting with “This complexity may involve” a typo and/pr  and/or

2. Page 4 starting with “The work presented herein”. As “relevance of canine to human cancer” seems confusing, perhaps, use “relevance of canine-to-human cancer studies” instead?

3. Page 5. Capitalize P-values at the beginning of the sentence (“P-values associated.. “)

4. Page 11 starting with “Most of these targets”” “are involved the process” is missing “in”

Figures

1. Figure 2.

a. Since red/green color scheme is not colorblind-friendly, consider changing to the previously suggested red/blue.

b. Instead of showing g1,g2,g3 for each module, show some of the real gene names that you mention in the discussion section.

c. For the consistency, please reorder as in Figure 3 PULM>MEL>OSA etc

2. Figure 3.

a. Please add module size to the x axis labels as BLSA (M4) N=…genes

b. Please label some at least some of the “outlier observations that are of interest as genes important ton cancers”. These are indeed the most interesting candidates, and the readers will benefit from knowing how the corresponding expression levels are distributed.

c. Y axis label. Please add units (Z scores)

**Have the authors made all data and (if applicable) computational code underlying the findings in their manuscript fully available?**

Reviewer #5: Yes

Reviewer #6: Yes

Reviewer #7: None

PLOS authors have the option to publish the peer review history of their article (what does this mean?). If published, this will include your full peer review and any attached files.

Reviewer #5: No

Reviewer #6: No

Reviewer #7: No

Figure Files:

Data Requirements:

Reproducibility:

References:

---

## [Editor Report · Decision Letter 3]

14 Sep 2021

Dear Dr. Tawa,

We are pleased to inform you that your manuscript 'Transcriptomic profiling in canines and humans reveals cancer specific gene modules and biological mechanisms common to both species' has been provisionally accepted for publication in PLOS Computational Biology.

Best regards,

James Gallo

Associate Editor

PLOS Computational Biology

Florian Markowetz

Deputy Editor

PLOS Computational Biology

---

## [Editor Report · Acceptance letter]

23 Sep 2021

PCOMPBIOL-D-21-00257R3 

 Transcriptomic profiling in canines and humans reveals cancer specific gene modules and biological mechanisms common to both species 

Dear Dr Tawa,

I am pleased to inform you that your manuscript has been formally accepted for publication in PLOS Computational Biology. Your manuscript is now with our production department and you will be notified of the publication date in due course.

With kind regards,

Andrea Szabo
